# ENHANCING TIME-SERIES FORECASTING WITH ITERATIVE DECOMPOSITION AND SEPARABLE TRAINING

## ABSTRACT

Time series data, crucial for decision-making in fields like finance and healthcare, often presents challenges due to its inherent complexity, exacerbating the bias-variance tradeoff and leading to overfitting and underfitting in conventional forecasting models. While promising, state-of-the-art models like PatchTST, iTransformer, and DLinear are hindered by this tradeoff, limiting their ability to separate predictable patterns from noise. To resolve this, we propose the IDEAS framework, which reduces the bias-variance tradeoff to help models achieve optimal performance. IDEAS combines iterative residual decomposition, which reduces bias by extracting predictable patterns, and separable training, which reduces variance by independently optimizing each component. We provide theoretical proof and demonstrate through experiments that IDEAS significantly improves performance across four state-of-the-art models on nine complex benchmark datasets, offering a more robust solution for complex time series forecasting.

## 1 INTRODUCTION

Time series data plays an essential role in various fields such as weather forecasting, medical diagnosis, and traffic prediction. It is widely used to solve numerous real-world problems and has a significant impact on our daily lives (Esling & Agon, 2012; Shumway et al., 2000). Due to the importance and necessity of time series research, numerous studies have been proposed in recent years for time series forecasting. In particular, unlike conventional prediction-focused studies, recent research has aimed to analyze and predict time series data from various perspectives. For instance, PatchTST (Nie et al., 2022) and iTransformer (Liu et al., 2023) focus on segmenting time series data along the temporal axis or emphasize variable-centered learning. On the other hand, models like DLinear (Zeng et al., 2023) and TimeMixer (Wang et al., 2024) aim to enhance learning by decomposing time series data into individual components.

Despite these advancements, time series data remains particularly challenging due to its inherent complexity, which is distinct from other types of sequential data like language or video. Time series data consists of multiple intertwined components, including long-term trends, seasonal patterns, and cycles, as well as unpredictable noise that often complicates the learning process. Unlike other types of data, these components must be separated to accurately model the underlying patterns. This mixture of predictable and unpredictable patterns often leads to a bias-variance tradeoff, where models either overfit to noise or underfit by failing to capture meaningful patterns (Ramasubramanian, 2007; Chen et al., 2014). Conventional forecasting models attempt to learn all components of time series data—both meaningful patterns and random noise—simultaneously. This approach often leads to overfitting, where models capture noise as if it were signal, or underfitting, missing important patterns (cf.Figure. 1). As a result, the models fail to generalize well on new data. Noise embedded within the data exacerbates the problem, causing overfitting when models fail to effectively distinguish between noise and meaningful patterns (Ying, 2019). Even when noise is excluded, the simultaneous learning of multiple components can still result in sub-optimal learning due to the model's inability to focus on each component's unique characteristics. These limitations emphasize the need for a more fundamental solution that directly addresses the complexity and inherent bias-variance tradeoff in time series data, rather than just optimizing model performance.

To address the challenges of the bias-variance tradeoff in time series forecasting, we propose a novel and effective framework called **I**terative residual **DE**composition **A**nd **S**eperable training (IDEAS).

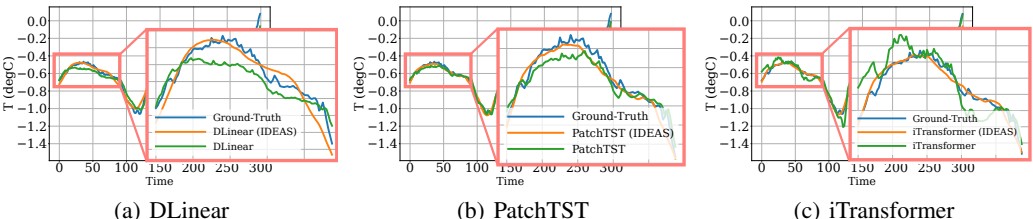

(a) DLinear          (b) PatchTST          (c) iTransformer

Figure 1: Visualization results of state-of-the-art models, DLinear, PatchTST, and iTransformer on the Weather dataset. Each model achieves impressive MSE values of $0.176$, $0.149$, and $0.174$, respectively. When visualized over a long sequence, all models closely follow the ground truth. However, as highlighted in the zoomed-in pink box, there are noticeable discrepancies between the predictions (green line) and the ground truth (blue line). Applying IDEAS (orange line) reduces these gaps, resulting in predictions that align more closely with the ground truth.

This comprehensive framework provides a robust and systematic solution by directly tackling the inherent bias-variance tradeoff through two key and complementary components: (i) iterative residual decomposition, which primarily reduces bias and confidently acts as an unbiased estimator, and (ii) separable training, which effectively minimizes variance. The IDEAS framework progressively handles the inherent complexity of time series data by isolating the predictable patterns from unpredictable noise, and ultimately enabling more efficient learning.

The first component, iterative residual decomposition, is specifically designed as an unbiased estimator that iteratively and systematically separates predictable patterns from unpredictable noise in a detailed, step-by-step manner. By consistently treating the residuals as approximate white noise in the theoretical limit of infinite iterations, this robust approach ensures that the model captures all significant and meaningful patterns without inadvertently introducing any bias, gradually and consistently enhancing prediction accuracy. This process allows for a clearer and more accurate extraction of the underlying structure of the time series, ultimately leading to more reliable and confident forecasting. The second component, separable training, directly addresses variance by training each decomposed component independently and distinctly. This independent training strategy effectively minimizes the overall impact of residual noise during the training process, further reducing the risk of potential overfitting and significantly enhancing the model's overall generalization ability.

By combining these two components, our approach achieves an optimal balance of the bias-variance tradeoff, resulting in more accurate and reliable predictions, even with complex and noisy datasets. Moreover, the IDEAS framework is highly adaptable and can be integrated with various existing time series decomposition methods and forecasting models, making it applicable across a wide range of fields, including finance, meteorology, and healthcare. Our main contributions of IDEAS are as follows:

1. We propose a novel method, iterative residual decomposition, which iteratively decomposes time series data into multiple predictable patterns and unpredictable patterns (noise). By progressively focusing on predictable components, the method effectively reduces bias and enhances the model's ability to capture the true underlying patterns in the data.

2. The predictable patterns obtained from the iterative residual decomposition are then trained independently, using separable training to reduce variance. We support this approach with mathematical analysis that demonstrates how separable training minimizes the influence of noise and overcomes the bias-variance tradeoff, leading to more efficient learning.

3. Our proposed framework, IDEAS, is versatile and can be applied to a wide range of time series decomposition algorithms and forecasting models. We validate the effectiveness of IDEAS through experiments on 9 datasets and 4 state-of-the-art models, achieving significant performance improvements.

These contributions represent a significant step forward in addressing the fundamental bias-variance tradeoff and the complexity of noise present in time series data. By focusing on this core challenge, IDEAS improves model performance and enhances the overall ability to generalize across various forecasting tasks, offering a more robust and accurate solution for time series forecasting.

## 2 RELATED WORKS

### 2.1 BIAS-VARIANCE TRADEOFF IN TIME SERIES FORECASTING MODEL

In time series forecasting, the bias-variance tradeoff is a fundamental challenge, especially with the increasing complexity of datasets. Models with high bias tend to underfit, missing critical patterns in the data, while models with high variance overfit, capturing noise along with the true signal. This tradeoff becomes even more pronounced in time series data due to temporal dependencies, noise, and dynamic patterns (Geman et al., 1992; Baek & Kim, 2018). As the data's complexity grows, the difficulty of finding a model that appropriately balances bias and variance increases, making it crucial to develop advanced techniques to handle these challenges.

Although recent approaches such as Transformer-based models have reduced overfitting to some extent, they often fail to balance the bias-variance tradeoff in more complex or noisy datasets. To mitigate this, our proposed method integrates the iterative residual decomposition and separable training. By isolating noise and training components separately, we aim to minimize overfitting while ensuring the model captures essential time-dependent patterns, thus addressing the bias-variance tradeoff more effectively. This balanced approach not only enhances model performance but also ensures that predictions are more reliable, even in the presence of intricate temporal patterns.

### 2.2 TIME SERIES DECOMPOSITION

Conventional time series decomposition methods, like STL or STR (Wen et al., 2019), break down data into trend, seasonality, and residual components to simplify analysis and forecasting. While effective in capturing patterns, these methods struggle with the growing complexity and non-linearity of modern datasets. As the scale and diversity of data increase, traditional decompositions may not adapt well, leading to loss of critical information in the residuals. Residuals, often treated as noise, can still contain valuable and predictive information. Research shows that residuals with autocorrelation or non-zero mean suggest models have missed capturing important underlying patterns(Dama & Sinoquet, 2021; Mauricio, 2008), underscoring the urgent need for improved and more advanced decomposition techniques. Therefore, enhancing decomposition methods is essential to extract more meaningful and actionable insights from increasingly complex time series data.

To address these issues, we propose the iterative residual decomposition method, which iteratively applies decomposition to residuals. This helps separate predictable patterns from noise, reducing data complexity and improving forecasting accuracy where conventional methods fall short.

### 2.3 TIME SERIES FORECASTING MODELS

Recent models such as Autoformer, DLinear (Zeng et al., 2023), and TimeMixer (Wang et al., 2024) employ decomposition techniques to capture underlying patterns in complex time series data. At the same time, models like PatchTST (Nie et al., 2022) and iTransformer (Liu et al., 2023) emphasize variable-centered learning and temporal segmentation to handle multivariate time series. These approaches have led to significant improvements in managing different types of time series data, highlighting the importance of decomposition and segmentation in improving forecasting accuracy.

However, a persistent challenge is overfitting. Although these models successfully capture intricate patterns, they often struggle to fully differentiate between meaningful signals and noise, which diminishes their ability to generalize effectively in real-world scenarios. This issue is particularly prevalent in datasets with high variability and noise, where models inadvertently learn irrelevant fluctuations alongside actual patterns, ultimately degrading their performance.

To address these challenges, we propose a novel approach that combines iterative residual decomposition with separable learning. Iterative residual decomposition progressively separates predictable patterns from residual noise, systematically reducing bias in the bias-variance tradeoff. The combination of these techniques not only resolves the overfitting problem but also offers a scalable solution to manage the increased complexity and variability inherent in modern time series data.

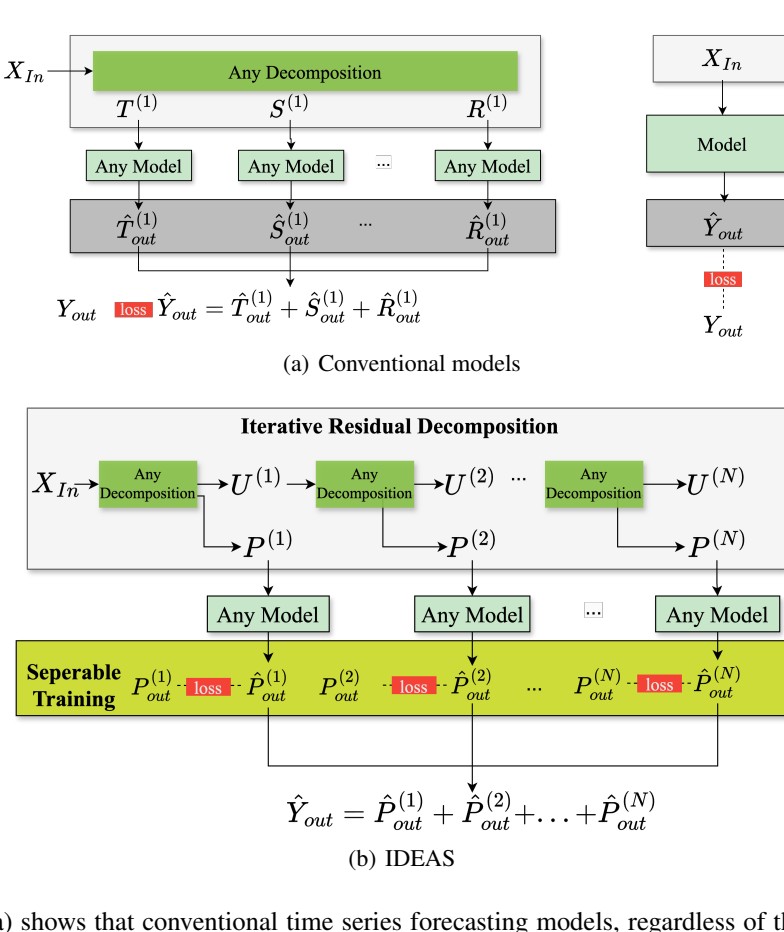

(a) Conventional models

(b) IDEAS

Figure 2: (a) shows that conventional time series forecasting models, regardless of their structure, aim to minimize the loss between the unified prediction $\hat{Y}_{out}$ and $Y_{out}$, treating the time series as a single entity. In contrast, (b) illustrates how separate learning is applied to each component, which helps prevent overfitting and underfitting by optimizing each part individually.

## 3 PROPOSED METHODS

Despite recent significant advancements in time series forecasting models, the inherent complexity of time series data continues to make the mitigation of overfitting, driven by the bias-variance trade-off (Assandri et al., 2023; Baidya & Lee, 2024), a challenging issue. In this section, we introduce a novel method, IDEAS, designed to address this problem. We first present the overall workflow of IDEAS, followed by a detailed description of its two main architectures with theoretical proofs. The detailed learning algorithm is given in the Appendix F, and the algorithm for iterative residual decomposition method is given in Algorithm 1.

### 3.1 OVERALL WORKFLOW

Figure 2 (b) shows the detailed design of our method, IDEAS. The overall workflow is as follows:

1. The input sequence $X_{In}$ is decomposed into unpredictable patterns $U^{(i)}$ (such as residuals) and predictable patterns $P^{(i)}$ using any time series decomposition method.

2. We iteratively apply the time series decomposition to the unpredictable pattern $U^{(i)}$ to obtain the noise-like unpredictable pattern $U^{(N)}$ and the meaningful predictable patterns $P^{(i)}$ obtained at each step $i$.

3. In our proposed separable training, the predictable patterns $\{P^{(i)}\}_{i=1}^{N}$ obtained from the iterative residual decomposition is used to train, at each step $i$, any arbitrary model (such as DLinear, PatchTST, etc.) to predict future value $P_{out}^{(i)}$.

---

**Algorithm 1:** Iterative residual decomposition method

---

**Input:** Input time series data $X_{In}$, Any forecasting model $\theta_f$, residual iteration number $max\_iter = N$

1  $i \leftarrow 1$;
2  Decompose $X_{In}$ into $P^{(1)}$ and $U^{(1)}$;
3  $i \leftarrow 2$;
4  **while** $i < max\_iter$ **do**
5  $\quad$ Decompose $U^{(i-1)}$ into $P^{(i)}$ and $U^{(i)}$;
$\quad\quad$ /* Can be use various decomposition methods(STR,STL,etc.)  */
6  $\quad$ $i \leftarrow i + 1$;
7  **return** $\{\hat{P}^{(i)}\}_{i=1}^{N}$

---

    4. During this separable training procedure, the loss function (which is in red box) is defined for each step $i$, allowing for individual learning at each step $i$.

    5. The prediction $\hat{Y}_{out}$ is obtained by summing the predicted outputs $\{\hat{P}_{out}^{(i)}\}_{i=1}^{N}$.

## 3.2 ITERATIVE RESIDUAL DECOMPOSITION

To effectively address the bias-variance tradeoff in time series forecasting, we propose an iterative residual decomposition method that iteratively extracts predictable patterns while reducing the impact of noise throughout the training process. This approach enhances the model's ability to generalize by separating meaningful patterns from noise at each step, thereby reducing the risk of overfitting or underfitting as demonstrated in Theorem 1 and Theorem 2.

Traditional decomposition methods, such as STL or STR (Dokumentov et al., 2015; Hyndman & Athanasopoulos, 2018), often leave residuals containing overlooked predictable patterns, which can still contribute to the bias-variance tradeoff. Our iterative residual decomposition method addresses this by repeatedly applying a decomposition process to extract these overlooked patterns, refining the residuals over multiple iterations. This systematic extraction ensures that residuals converge toward white noise, functioning as an unbiased estimator in the limit of infinite iterations. Consequently, our approach effectively separates predictable patterns from unpredictable patterns and mitigates the inherent bias introduced when models fail to capture all relevant patterns during training, resulting in improved forecasting accuracy.

Figure 3 clearly demonstrates how the iterative decomposition progressively refines the residuals towards a Gaussian distribution, validating the effectiveness of our method. In the iterative residual decomposition, the initial decomposition of the input time series data $X_{In}$ follows standard methods like STL or STR, which separate the time series into predictable components $P^{(1)}$ (e.g., trend, seasonality) and unpredictable residuals $U^{(1)}$ (e.g., noise):

$$U^{(1)} = X_{In} - P^{(1)}, \tag{1}$$

At each subsequent step, the residual $U^{(N-1)}$ is further decomposed to extract any remaining predictable patterns $P^{(N)}$, progressively refining the residuals:

$$U^{(N)} = U^{(N-1)} - P^{(N)}, \tag{2}$$

where $P^{(N)}$ represents the $N$-th set of predictable patterns extracted during the decomposition process. For instance, in the first decomposition step, $P^{(1)}$ may capture large-scale patterns like trends, while in later steps, $P^{(N)}$ captures more subtle predictable patterns, leaving the residuals $U^{(N)}$ to approach random noise as more predictable patterns are removed. As observed in Figure 3, the residuals (blue bars) progressively become more similar to a normal distribution (red dashed line) over successive iterations, confirming the iterative refinement of the residuals toward a more Gaussian-like distribution. As the number of iterations $N$ increases and approaches infinity ($N \to \infty$), the residuals $U^{(N)}$ are expected to converge to pure white noise, characterized by zero mean, constant variance, and no autocorrelation (cf. Figure 4). This convergence demonstrates that the iterative residual decomposition method ultimately yields an unbiased estimation of the predictable components, effectively separating the signal from the noise in the time series data.

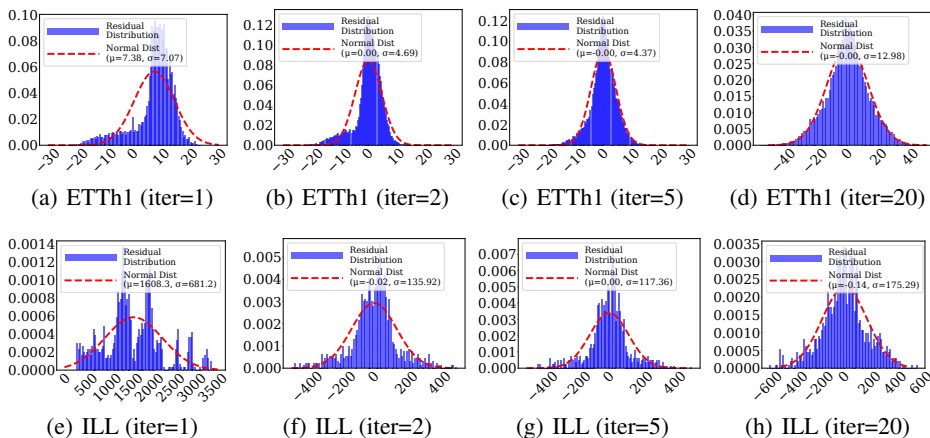

Figure 3: Visualization of the residual distributions over successive iterations for the ETTh1 and ILL datasets. As the iteration process progresses, the residuals (blue bars) increasingly resemble a normal distribution (red dashed line), indicating that STL decomposition method iteratively refines the residuals towards a more Gaussian-like distribution. More figures are in Appendix H.1

### 3.2.1 UNBIASEDNESS OF ITERATIVE RESIDUAL DECOMPOSITION

We assume that residuals will eventually converge to noise if each iteration successfully removes predictable patterns, even in complex datasets with nonlinearity or non-stationarity. This assumption is consistent with traditional methods like STR and STL, which are designed to handle non-stationary time series, making it a mild assumption.

To empirically validate this assumption, we propose using statistical tests such as the Shapiro-Wilk test for normality and autocorrelation tests (e.g., ACF or Durbin-Watson) to assess whether the residuals exhibit characteristics of noise after each iteration, as shown in Figure 4.

**Theorem 1** (Unbiasedness of iterative residual decomposition). *Let $X_{In}$ be a non-stationary time series. Assume $X_{in}$ can be iteratively decomposed into predictable components $P^{(i)}$ and residual components $U^{(i)}$. As the number of iterations $N \to \infty$, the expected value of the final residuals $U^{(N)}$ will converge to white noise, indicating the iterative residual decomposition process yields an asymptotically unbiased estimator of the predictable components of $X_{In}$.*

**Remark:** In the limit as $N \to \infty$, the iterative residual decomposition process is an unbiased estimator of the predictable components of $X_{In}$. Practically, we observe that with a finite number of iterations, $U^{(N)}$ closely approximates white noise, demonstrating consistent evidence of near-unbiasedness across diverse real-world applications and datasets.

The iterative residual decomposition not only theoretically proves unbiasedness but also empirically validates this through real-world testing. The theoretical proof of Theorem 1 is provided in Appendix A. To complement this theoretical analysis, we empirically validate Theorem 1 by conducting ACF and Durbin-Watson tests across multiple iterations of the iterative residual decomposition. As shown in Figure 4, the Durbin-Watson statistic (blue line) approaches the ideal value of 2.0 (red dashed line), and the ACF statistic (orange line) converges towards 0.0 (pink dashed line) as the number of iterations increases. This convergence indicates that the residuals become progressively closer to white noise, supporting our theorem. More visualization of Figure 4 are in Appendix H.2.

## 3.3 SEPARABLE TRAINING

To address the limitations of conventional training methods in time series forecasting, we propose a separable training approach that optimizes each component individually, thus enhancing the model's learning effectiveness and overall forecasting accuracy. Unlike conventional methods that train on all components simultaneously, our approach ensures that each component—such as trend, seasonality, and cycles—is trained separately, allowing the model to capture the distinct information contained within each component more effectively.

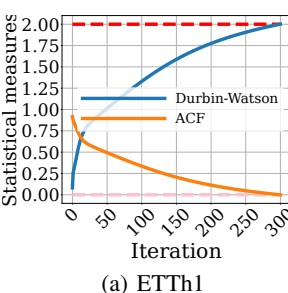 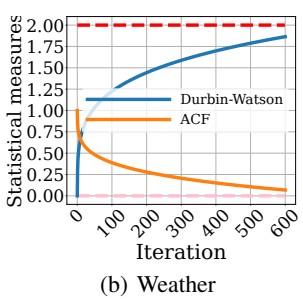 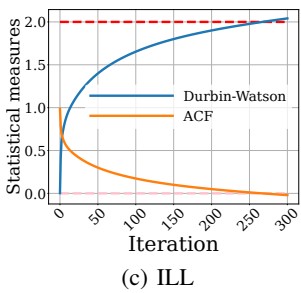

|     (a) ETTh1     |     (b) Weather     |     (c) ILL     |

Figure 4: Empirical validation showing that as iterations increase, the residuals approach white noise, demonstrated by the Durbin-Watson and ACF statistics converging towards their expectation.

Time series data consists of various components, such as trend, seasonality, and cycles, each carrying distinct information (Baidya & Lee, 2024). Unlike image or language data, where components are often treated as a whole, each component in time series data has a specific meaning. For example, the trend captures long-term movements, while seasonality highlights periodic fluctuations.

Conventional forecasting models often lead to sub-optimal performance or overfitting by training on all these components simultaneously (Assandri et al., 2023; Gelman & Hill, 2007; Liu & Wang, 2024). In contrast, separable training allows the model to optimize each component independently, avoiding these issues. Rather than introducing a completely new approach, separable training generalizes and extends the conventional methods by allowing the model to optimize for each component separately, thus enhancing learning effectiveness for each aspect of the data.

With separable training, the model can focus on learning each component more effectively, which leads to improved forecasting performance, as demonstrated in our experiments. Mathematically, we independently optimize each component $P_{out}^{(1)}, P_{out}^{(2)}, \ldots, P_{out}^{(N)}$, ensuring that the training process for each component is more tailored and precise, as shown below:

$$\theta_{P_{out}^{(1)}} := \theta_{P_{out}^{(1)}} - \eta \left( \frac{2}{T} \sum_{t=1}^{T} (P_{out,t}^{(1)} - \hat{P}_{out,t}^{(1)}(\theta_{P_{out}^{(1)}})) \nabla_{\theta_{P_{out}^{(1)}}} \hat{P}_{out,t}^{(1)}(\theta_{P_{out}^{(1)}}) \right),$$

$$\theta_{P_{out}^{(2)}} := \theta_{P_{out}^{(2)}} - \eta \left( \frac{2}{T} \sum_{t=1}^{T} (P_{out,t}^{(2)} - \hat{P}_{out,t}^{(2)}(\theta_{P_{out}^{(2)}})) \nabla_{\theta_{P_{out}^{(2)}}} \hat{P}_{out,t}^{(2)}(\theta_{P_{out}^{(2)}}) \right), \quad (3)$$

$$\cdots$$

$$\theta_{P_{out}^{(N)}} := \theta_{P_{out}^{(N)}} - \eta \left( \frac{2}{T} \sum_{t=1}^{T} (P_{out,t}^{(N)} - \hat{P}_{out,t}^{(N)}(\theta_{P_{out}^{(N)}})) \nabla_{\theta_{P_{out}^{(N)}}} \hat{P}_{out,t}^{(N)}(\theta_{P_{out}^{(N)}}) \right)$$

In Theorem.2, we demonstrate how separable training improves the model's ability to handle complex time series data by reducing the overall model variance and enhancing predictive accuracy. This approach helps the model achieve better overall performance by avoiding the limitations of unified training approaches. By learning each component independently, the model can avoid overfitting to dominant components or noise, resulting in a more robust and accurate forecasting process.

**Theorem 2** (Bias-Variance Tradeoff improvement with separable training). *Let a time series $X_{In}$ be decomposed into $N$ components $P^{(1)}, P^{(2)}, \ldots, P^{(N)}$, each representing distinct patterns. Separable training optimizes each component individually, reducing the overall model variance by eliminating the covariance between components, which leads to improved forecasting performance.*

**Remark:** The separable training paradigm decouples the interactions between components, significantly reducing the overall variance $\sigma_{X_{In}}^2$. By removing the covariance terms between components, the model becomes more resilient to overfitting, improving its generalization capability to unseen data. This decoupling mechanism addresses the bias-variance tradeoff, enabling the model to achieve higher predictive accuracy and overall performance in time series forecasting.

### 3.3.1 BETTER BIAS-VARIANCE TRADEOFF

A significant advantage of separable training is its ability to effectively manage the bias-variance tradeoff, a crucial factor in balancing model complexity and generalization. By training each component independently, separable training reduces variance while maintaining low bias, resulting in more accurate and generalized forecasting.

In conventional unified training, all components of the time series—trend, seasonality, and noise—are optimized together. This introduces significant variance, as the model attempts to fit all components at once, often leading to overfitting to noisy patterns and underfitting of subtle trends.

By separating the training process, separable training eliminates the covariance between unpredictable and predictable patterns, leading to a reduced overall variance. This independent training process ensures that each component is optimized more effectively, contributing to a more balanced and accurate forecasting model. Mathematically, the variance in a conventional model can be expressed as:

$$\sigma^2_{X_{In}} = \sigma^2_{P^{(1)}_{out}} + \sigma^2_{P^{(2)}_{out}} + \cdots \sigma^2_{P^{(N)}_{out}} + \sigma^2_{U^{(N)}} + 2 \cdot \Sigma(P^{(1)}_{out}, P^{(2)}_{out}, \cdots, P^{(N)}_{out}, U^{(N)}) \qquad (4)$$

The covariance term, $2 \cdot \Sigma(P^{(1)}_{out}, P^{(2)}_{out}, \cdots, P^{(N)}_{out}, U^{(N)})$, represents the interaction between the noise and the meaningful components, which can increase the overall variance and lead to overfitting.

By separating the training process, separable training effectively removes these covariance terms, leading to a reduction in the model's overall variance:

$$\sigma^2_{X_{In}} = \sigma^2_{P^{(1)}_{out}} + \sigma^2_{P^{(2)}_{out}} + \cdots \sigma^2_{P^{(N)}_{out}} \qquad (5)$$

This reduction in variance helps mitigate the risk of overfitting, while the independent optimization of each component maintains low bias. As a result, separable training effectively balances the bias-variance tradeoff, leading to improved generalization and forecasting performance compared to conventional unified training methods.

## 4 EXPERIMENTS

In this section, we present the experimental results of IDEAS on mainstream LTSF benchmarks. We also discuss the efficiency advantages introduced by the proposed IDEAS method. We conduct ablation studies and analyses to further demonstrate the effectiveness of the IDEAS approach.

### 4.1 EXPERIMENTAL SETTINGS

We list all the descriptions of datasets, baselines and detailed experimental settings in Appendix D.

**Datasets:** We conducted experiments on 9 mainstream LTSF datasets, including Weather, Exchange rate, ETTh1, ETTh2, ETTm1, ETTm2, Electricity, Traffic, and National illness (ILL). The details of these datasets are presented in Appendix D.2. Additionally, we conducted stationarity testing on each benchmarked dataset using time series stationarity tests, including the Augmented Dickey-Fuller (ADF) test (Mushtaq, 2011) and the Kwiatkowski-Phillips-Schmidt-Shin (KPSS) test (Baum, 2018). The tests were performed for each variable, and the overall characteristics of stationarity were documented accordingly (detailed results of stationarity testing are in Appendix C). Based on the statistical test results, we classify the datasets into non-stationary (Weather, Exchange, ETTh1, ETTm1) and weak non-stationary (ETTh2, ETTm2, Electricity, Traffic, ILL) categories.

### 4.2 MAIN RESULTS

In particular, non-stationary time series data are more complex due to evolving patterns, such as changing trends and seasonality. The IDEAS framework is designed to handle such complexity, making it highly effective for real-world time series data where non-stationarity is prevalent.

Table 1: Stationarity testing results on benchmarked dataset. X indicates that both ADF and KPSS tests consider non-stationary, while △ denotes that at least one of the test consider it non-stationary. We refer to cases marked with △ as weak non-stationary.

|  | Weather | Exchange | ETTh1, ETTm1 | ETTh2, ETTm2 | Electricity | Traffic | ILL |
|---|---|---|---|---|---|---|---|
| Stationarity | X | X | X | △ | △ | △ | △ |

Table 2: Experimental results on 9 benchmark datasets. Results where the IDEAS method improved the performance of a base model are highlighted in **bold**, and the best performance w.r.t. a pair of (dataset, horizon $h$) among all is marked in **blue**. $h$ refers to the prediction horizon (or length).

| Datasets | h | Weather MSE | Weather MAE | Exchange MSE | Exchange MAE | ETTh1 MSE | ETTh1 MAE | ETTm1 MSE | ETTm1 MAE | ETTh2 MSE | ETTh2 MAE | ETTm2 MSE | ETTm2 MAE | Electricity MSE | Electricity MAE | Traffic MSE | Traffic MAE | ILL h | ILL MSE | ILL MAE |
|---|---|---|---|---|---|---|---|---|---|---|---|---|---|---|---|---|---|---|---|---|
| DLinear (2023) | 96 | 0.176 | 0.237 | 0.081 | 0.203 | 0.375 | 0.399 | 0.299 | 0.343 | 0.289 | 0.353 | 0.167 | 0.260 | 0.140 | 0.237 | 0.410 | 0.282 | 24 | 2.215 | 1.081 |
|  | 192 | 0.220 | 0.282 | 0.157 | 0.293 | 0.405 | 0.416 | 0.335 | 0.365 | 0.383 | 0.418 | 0.224 | 0.303 | 0.153 | 0.249 | 0.423 | 0.287 | 36 | 1.963 | 0.963 |
|  | 336 | 0.265 | 0.319 | 0.305 | 0.414 | 0.439 | 0.443 | 0.369 | 0.386 | 0.448 | 0.465 | 0.281 | 0.342 | 0.169 | 0.267 | 0.436 | 0.296 | 48 | 2.130 | 1.024 |
|  | 720 | 0.323 | 0.362 | 0.643 | 0.601 | 0.472 | 0.490 | 0.425 | 0.421 | 0.605 | 0.551 | 0.397 | 0.421 | 0.203 | 0.301 | 0.466 | 0.315 | 60 | 2.368 | 1.096 |
| IDEAS DLinear | 96 | **0.143** | **0.206** | **0.073** | **0.194** | **0.240** | **0.334** | **0.245** | **0.318** | **0.270** | **0.336** | **0.144** | **0.241** | **0.123** | **0.227** | **0.339** | **0.281** | 24 | **1.776** | **0.956** |
|  | 192 | **0.199** | **0.266** | 0.159 | **0.293** | **0.314** | **0.377** | **0.283** | **0.338** | **0.342** | **0.395** | **0.222** | **0.302** | **0.142** | **0.246** | **0.397** | **0.298** | 36 | **1.374** | **0.801** |
|  | 336 | **0.255** | **0.315** | **0.304** | **0.413** | **0.359** | **0.420** | **0.317** | **0.360** | **0.427** | **0.457** | **0.270** | **0.339** | **0.158** | **0.263** | **0.435** | **0.315** | 48 | **1.415** | **0.825** |
|  | 720 | **0.320** | **0.367** | **0.632** | **0.635** | **0.385** | **0.449** | **0.405** | **0.418** | 0.612 | **0.554** | 0.398 | 0.421 | **0.194** | **0.296** | **0.457** | 0.317 | 60 | **1.585** | **0.878** |
| Imp.(Avg.) |  | 8.25% | 4.66% | 2.66% | -0.24% | 23.8% | 9.81% | 13.1% | 5.53% | 5.20% | 2.87% | 4.58% | 2.13% | 7.57% | 2.15% | 2.15% | -2.63% |  | 29.1% | 16.9% |
| PatchTST (2023) | 96 | 0.149 | 0.198 | 0.093 | 0.218 | 0.370 | 0.400 | 0.290 | 0.342 | 0.274 | 0.336 | 0.165 | 0.255 | 0.129 | 0.222 | 0.360 | 0.249 | 24 | 1.319 | 0.754 |
|  | 192 | 0.194 | 0.241 | 0.208 | 0.332 | 0.413 | 0.429 | 0.332 | 0.369 | 0.339 | 0.379 | 0.220 | 0.292 | 0.147 | 0.240 | 0.379 | 0.256 | 36 | **1.007** | 0.870 |
|  | 336 | 0.245 | 0.282 | 0.359 | 0.440 | 0.422 | 0.440 | 0.366 | 0.392 | 0.384 | 0.384 | 0.274 | 0.329 | 0.163 | 0.259 | 0.392 | 0.264 | 48 | 1.553 | 0.815 |
|  | 720 | 0.314 | 0.334 | 1.194 | 0.815 | 0.447 | 0.468 | 0.416 | 0.420 | 0.379 | 0.422 | 0.362 | 0.385 | 0.197 | 0.290 | 0.432 | 0.286 | 60 | **1.016** | 0.788 |
| IDEAS PatchTST | 96 | **0.128** | **0.182** | **0.066** | **0.176** | **0.264** | **0.343** | **0.240** | **0.311** | **0.260** | **0.322** | **0.163** | **0.251** | **0.115** | **0.217** | **0.358** | **0.247** | 24 | **1.063** | **0.706** |
|  | 192 | **0.178** | **0.225** | **0.165** | **0.287** | **0.303** | **0.368** | **0.282** | **0.339** | **0.338** | **0.375** | 0.225 | 0.295 | **0.143** | **0.234** | **0.377** | **0.254** | 36 | 1.118 | 0.728 |
|  | 336 | **0.235** | **0.273** | 0.328 | **0.415** | **0.334** | **0.400** | **0.308** | **0.359** | **0.327** | **0.377** | 0.272 | **0.327** | **0.158** | **0.249** | 0.392 | **0.263** | 48 | 1.298 | 0.755 |
|  | 720 | **0.301** | **0.325** | 0.875 | 0.695 | **0.348** | **0.416** | **0.354** | **0.388** | **0.378** | **0.418** | 0.364 | **0.384** | **0.192** | **0.287** | 0.437 | **0.288** | 60 | 1.260 | 0.753 |
| Imp.(Avg.) |  | 7.64% | 5.15% | 21.3% | 13.3% | 24.6% | 12.2% | 15.8% | 8.31% | 1.57% | 1.99% | -0.22% | 0.35% | 4.79% | 2.41% | 0.21% | 0.31% |  | 0.20% | 8.62% |
| iTransformer (2024) | 96 | 0.174 | 0.214 | 0.086 | 0.206 | 0.386 | 0.405 | 0.334 | 0.368 | 0.297 | 0.349 | 0.180 | 0.264 | 0.148 | 0.240 | 0.395 | 0.268 | 24 | 2.085 | 0.953 |
|  | 192 | 0.221 | 0.254 | 0.177 | 0.299 | 0.441 | 0.436 | 0.377 | 0.391 | 0.380 | 0.400 | 0.250 | 0.309 | 0.162 | 0.253 | 0.417 | 0.276 | 36 | 1.973 | 0.947 |
|  | 336 | 0.278 | 0.296 | 0.331 | 0.417 | 0.487 | 0.458 | 0.426 | 0.420 | 0.428 | 0.432 | 0.311 | 0.348 | 0.178 | 0.269 | 0.433 | 0.283 | 48 | 2.124 | 1.018 |
|  | 720 | 0.358 | 0.347 | 0.847 | 0.691 | 0.503 | 0.491 | 0.491 | 0.459 | 0.427 | 0.445 | 0.412 | 0.407 | 0.225 | 0.317 | 0.467 | 0.302 | 60 | 2.164 | 1.032 |
| IDEAS iTransformer | 96 | **0.139** | **0.187** | **0.071** | **0.183** | **0.292** | **0.359** | **0.269** | **0.343** | **0.295** | **0.345** | 0.182 | 0.265 | **0.141** | 0.242 | **0.317** | **0.260** | 24 | **1.790** | **0.910** |
|  | 192 | **0.194** | **0.238** | **0.168** | **0.286** | **0.329** | **0.387** | **0.325** | **0.380** | **0.376** | 0.401 | **0.248** | **0.308** | **0.160** | 0.261 | **0.383** | **0.271** | 36 | **1.548** | **0.863** |
|  | 336 | **0.249** | **0.280** | **0.326** | **0.412** | **0.364** | **0.424** | **0.348** | **0.395** | **0.408** | **0.427** | **0.309** | **0.344** | **0.174** | 0.270 | **0.415** | **0.282** | 48 | **1.579** | **0.865** |
|  | 720 | **0.329** | **0.339** | **0.724** | **0.645** | **0.395** | **0.451** | **0.390** | **0.419** | 0.430 | 0.450 | 0.412 | 0.408 | **0.205** | **0.302** | **0.466** | 0.311 | 60 | **1.600** | **0.874** |
| Imp.(Avg.) |  | 12.7% | 6.66% | 9.64% | 5.84% | 24.1% | 9.54% | 18.0% | 6.07% | 1.42% | 0.23% | 0.08% | 0.212% | 4.27% | 0.09% | 8.07% | 0.54% |  | 21.9% | 10.9% |
| TimeMixer (2024) | 96 | 0.163 | 0.209 | 0.093 | 0.212 | 0.375 | 0.400 | 0.320 | 0.357 | 0.289 | 0.341 | 0.175 | 0.258 | 0.153 | 0.247 | 0.462 | 0.285 | 24 | 1.469 | 0.798 |
|  | 192 | 0.208 | 0.250 | 0.174 | 0.297 | 0.429 | 0.421 | 0.361 | 0.381 | 0.372 | 0.392 | 0.237 | 0.299 | 0.166 | 0.256 | 0.473 | 0.296 | 36 | 1.890 | 0.867 |
|  | 336 | 0.251 | 0.287 | 0.349 | 0.426 | 0.458 | 0.458 | 0.390 | 0.404 | 0.386 | 0.414 | 0.298 | 0.340 | 0.185 | 0.277 | 0.498 | 0.296 | 48 | 1.885 | 0.924 |
|  | 720 | 0.339 | 0.341 | 1.065 | 0.770 | 0.498 | 0.482 | 0.454 | 0.441 | 0.412 | 0.434 | 0.391 | 0.396 | 0.225 | 0.310 | 0.506 | 0.313 | 60 | 1.955 | 0.980 |
| IDEAS TimeMixer | 96 | **0.148** | **0.194** | **0.087** | **0.209** | **0.255** | **0.341** | **0.248** | **0.323** | **0.278** | 0.349 | **0.151** | **0.247** | **0.114** | **0.214** | **0.337** | **0.285** | 24 | **1.317** | **0.762** |
|  | 192 | **0.200** | **0.243** | **0.172** | **0.294** | **0.329** | **0.382** | **0.282** | **0.348** | **0.345** | **0.384** | **0.236** | **0.298** | **0.150** | **0.246** | **0.399** | 0.308 | 36 | **1.062** | **0.690** |
|  | 336 | **0.242** | **0.278** | **0.346** | **0.423** | **0.333** | **0.391** | **0.317** | **0.372** | **0.298** | **0.342** | **0.298** | 0.342 | **0.169** | **0.262** | **0.428** | 0.314 | 48 | **1.151** | **0.747** |
|  | 720 | **0.334** | **0.340** | **0.946** | **0.724** | **0.385** | **0.434** | **0.372** | **0.404** | 0.413 | 0.434 | **0.380** | 0.398 | **0.212** | **0.304** | **0.438** | 0.320 | 60 | **1.427** | **0.801** |
| Imp.(Avg.) |  | 4.53% | 3.35% | 4.98% | 1.43% | 27.3% | 12.2% | 20.3% | 8.62% | 3.35% | 0.10% | 4.24% | 0.88% | 12.4% | 6.15% | 17.5% | -3.09% |  | 30.0% | 15.6% |

Table 2 compares the performance of 4 state-of-the-art models with and without the IDEAS method. As discussed in (Shao et al., 2023), the effectiveness of each model varies depending on the characteristics of the time series dataset. For instance, Transformer models perform better on stationary datasets with clear and stable patterns, while Linear models may suffer from underfitting in such cases. Conversely, Linear models excel on non-stationary datasets with unclear patterns and significant distribution shifts, where Transformer models are more prone to overfitting.

Experimental results show that IDEAS can significantly improve the performance of existing models by addressing the bias-variance trade-off. For the highly abnormal ETTh1 dataset, all four models achieve about 25% improvement. In contrast, for the weak non-stationary Traffic dataset, the overall improvement is less than 10%. For the weak non-sationary ILL dataset, most models improve by more than 20% except PatchTST, which achieves only a small gain of 0.2%, indicating that it has already achieved the optimal performance. Overall, IDEAS effectively alleviates the limitations of the bias-variance trade-off and achieves the optimal performance of existing models.

## 4.3 SENSITIVITY AND ABLATION STUDIES

**Various time-series decomposition with IDEAS:** Table 3 presents a sensitivity analysis of IDEAS performance based on different time series decomposition methods. We compare results

Table 3: Ablation studies on various time-series decomposition with IDEAS. The best results are in **bold** and the second best are underlined.

| decomposition method | ETTh1 | | | | ILL | | | |
| --- | --- | --- | --- | --- | --- | --- | --- | --- |
| | DLinear | PatchTST | iTransformer | TimeMixer | DLinear | PatchTST | iTransformer | TimeMixer |
| Original model | 0.375 | 0.370 | 0.386 | 0.375 | 2.215 | 1.319 | 2.085 | 1.469 |
| IDEAS (STR) | 0.317 | 0.313 | 0.339 | 0.339 | 1.886 | 1.255 | 2.150 | 1.496 |
| IDEAS (STL) | **0.240** | **0.264** | **0.292** | **0.255** | **1.776** | **1.063** | **1.790** | **1.317** |

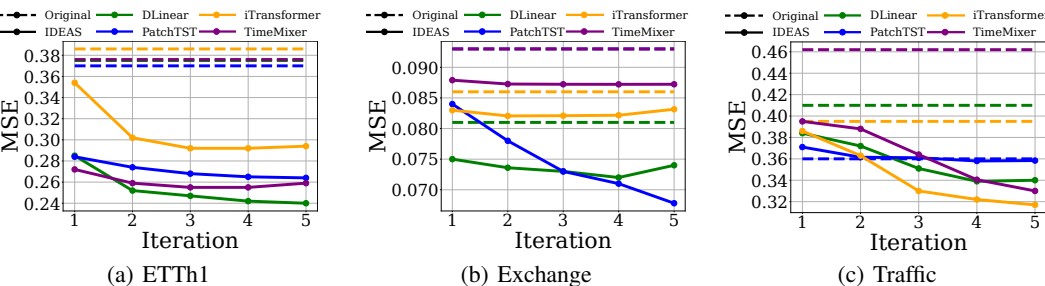

(a) ETTh1  (b) Exchange  (c) Traffic

Figure 5: Sensitivity to iteration number $N$ of separable training. More figures are in Appendix H.3.

on a non-stationary dataset (ETTh1) and a weak non-stationary dataset (ILL) to assess the effect of decomposition choice on forecasting accuracy. While the specific decomposition method influences the model's effectiveness, the application of IDEAS consistently enhances performance compared to the original models. In particular, for weak non-stationary datasets, the improvement achieved by IDEAS heavily depends on the selected decomposition technique, with certain methods yielding significant gains, while others may offer marginal or no improvements over the original models.

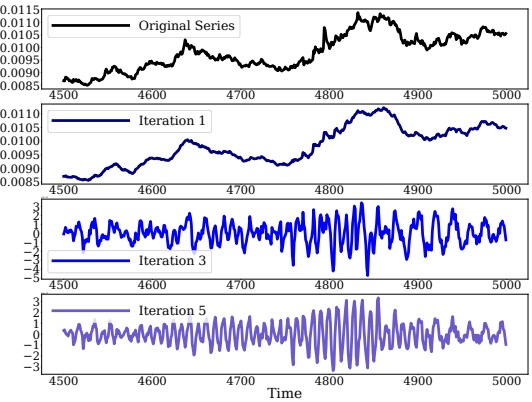

Figure 6: Visualization of the original time series (top) and the predictable patterns at each iteration of the iterative residual decomposition (from iteration 1 to 5). More figures are in Appendix H.4

**Sensitivity on separable training iterations:** Figure 5 demonstrates how MSE decreases progressively as the number of iterations in the iterative residual decomposition and separable training increases within the IDEAS framework. Across datasets like ETTh1, Exchange, and Traffic, repeated iterations consistently reduce MSE, improving forecasting accuracy compared to the original models (dotted line), highlighting the effectiveness of IDEAS. The framework iteratively learns from predictable patterns extracted at each decomposition stage, as shown in Figure 6, capturing the data's underlying structures. This process incrementally refines the model's ability to separate patterns from noise, addressing the bias-variance tradeoff more efficiently. The consistent reduction in MSE across iterations confirms IDEAS's robustness in enhancing time series forecasting.

## 5 CONCLUSION

In this paper, we introduce IDEAS, a novel framework that enhances time series forecasting by addressing the bias-variance tradeoff through iterative residual decomposition and separable learning. Our experiments demonstrate significant improvements on various benchmark datasets, particularly on non-stationary data where existing methods struggle with overfitting or underfitting. By separating predictable patterns from noise to reduce bias and minimizing variance, IDEAS achieves more accurate and robust forecasts, making it a promising solution for complex, noisy time series data. Future work could explore integrating IDEAS with other techniques and applying it to broader domains where accurate forecasting is essential.

**Ethics Statement** The IDEAS framework and associated time series forecasting models presented in this paper are designed to improve the accuracy and robustness of predictions in various domains, including finance, healthcare, and other sectors where decision-making based on time series data is critical. In applying these models, we acknowledge the importance of transparency, fairness, and accountability, especially when predictions have significant real-world consequences. Furthermore, no human subjects were involved in this research, and the datasets used are publicly available, ensuring compliance with relevant data privacy regulations.

**Reproducibility Statement** To ensure the reproducibility and completeness of this paper, we make our code available at `https://drive.google.com/drive/folders/1mIItRegJiPSLbixG_EogsjCsixLDQBYk?usp=drive_link`. We give details on our experimental protocol in the Appendix D.4.

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

## A   PROOF OF THEOREM 1

**Theorem 1** (Unbiasedness of iterative residual decomposition). *Let $X_{In}$ be a non-stationary time series. Assume $X_{In}$ can be iteratively decomposed into predictable components $P^{(i)}$ and residual components $U^{(i)}$. As the number of iterations $N \to \infty$, the expected value of the final residuals $U^{(N)}$ will converge to white noise, indicating that the iterative residual decomposition process yields an asymptotically unbiased estimator of the predictable components of $X_{In}$.*

*Proof.* The proposed iterative residual decomposition method is designed to iteratively extract and eliminate predictable patterns from the time series data, ultimately isolating pure noise as the iteration count approaches infinity. This proof establishes that the iterative residual decomposition serves as an asymptotically unbiased estimator of the predictable components and that the residuals progressively converge towards the properties of white noise.

We begin by decomposing the input time series $X_{In}$ using a standard decomposition method, such as STL or STR:

$$X_{In} = P^{(1)} + U^{(1)}, \tag{6}$$

where $P^{(1)}$ denotes the first extracted predictable pattern (e.g., trend, seasonality), and $U^{(1)}$ represents the initial residual component.

At each iteration $i$, the residual $U^{(i-1)}$ is further decomposed into:

$$U^{(i-1)} = P^{(i)} + U^{(i)}, \tag{7}$$

where $P^{(i)}$ captures the most predictable structure within $U^{(i-1)}$. This iterative process continues, with each subsequent iteration $i$ progressively removing additional predictable components from the residuals.

As the number of iterations $N \to \infty$, all predictable patterns are effectively eliminated, leading to residuals $U^{(N)}$ that comprise solely unpredictable elements or pure noise. In this asymptotic limit, the expectation of the residuals converges to zero:

$$\lim_{N \to \infty} \mathbb{E}[U^{(N)}] = 0. \tag{8}$$

This convergence implies that the residuals become entirely unbiased, signifying that the iterative residual decomposition method serves as an asymptotically unbiased estimator for the predictable components within $X_{In}$.

To further substantiate the convergence of residuals $U^{(N)}$ towards white noise, we consider the Central Limit Theorem (CLT), which states that the sum of a sufficiently large number of independent and identically distributed random variables approximates a normal distribution. We express the residual $U^{(N)}$ in terms of an aggregation of various random components:

$$U^{(N)} = \sum_{i=1}^{n} X_i(t), \tag{9}$$

where $X_i(t)$ denotes the individual random components at iteration $i$.

By invoking the CLT, as $n \to \infty$, the standardized sum of these components converges towards a normal distribution:

$$\lim_{n \to \infty} \frac{1}{\sqrt{n}} \sum_{i=1}^{n} X_i(t) \approx \mathcal{N}(0, \sigma^2). \tag{10}$$

This result confirms that the residuals $U^{(N)}$ progressively conform to a normal distribution with mean 0 and variance $\sigma^2$, thereby acquiring the attributes of white noise.

Additionally, the autocorrelation function (ACF) of the residuals $U^{(N)}$ at lag $h$ is defined as:

$$\rho_{U^{(N)}}(h) = \frac{\mathbb{E}[(U^{(N)}(t) - \mu_{U^{(N)}})(U^{(N)}(t+h) - \mu_{U^{(N)}})]}{\sigma^2_{U^{(N)}}}, \tag{11}$$

where $\mu_{U^{(N)}}$ and $\sigma_{U^{(N)}}^2$ are the mean and variance of $U^{(N)}$, respectively. As $N \to \infty$, since $\mu_{U^{(N)}} \approx 0$, it follows that:

$$\rho_{U^{(N)}}(h) \approx 0 \quad \text{for} \quad h \neq 0. \tag{12}$$

This demonstrates that the residuals $U^{(N)}$ become uncorrelated over iterations, thereby exhibiting the characteristics of white noise.

In conclusion, the iterative residual decomposition process not only acts as an asymptotically unbiased estimator but also ensures that residuals exhibit white noise properties as iterations increase. This theoretical foundation confirms the robustness and efficacy of the iterative residual decomposition method in managing the bias-variance tradeoff and enhancing the generalization capability of time series models in complex and noisy environments. $\qquad\square$

## B  PROOF OF THEOREM 2

**Theorem 2** (Bias-Variance Tradeoff improvement with separable training). *Let a time series $X_{In}$ be decomposed into $N$ components $P^{(1)}, P^{(2)}, \ldots, P^{(N)}$, each representing distinct patterns. Separable training optimizes each component individually and reduces the overall model variance by eliminating the covariance between components.*

*Proof.* In the conventional unified training paradigm, all components of the time series $X_{In}$ are trained simultaneously, resulting in an accumulation of both individual variances and covariance interactions among these components. The total variance of $X_{In}$ in this unified setting can be expressed as:

$$\sigma_{X_{In}}^2 = \sum_{i=1}^{N} \sigma_{P^{(i)}}^2 + 2 \sum_{1 \leq i < j \leq N} \text{Cov}(P^{(i)}, P^{(j)}), \tag{13}$$

where $\sigma_{P^{(i)}}^2$ denotes the variance of each individual component $P^{(i)}$, and $\text{Cov}(P^{(i)}, P^{(j)})$ represents the covariance between different components $P^{(i)}$ and $P^{(j)}$.

In unified training, the covariance term $2 \sum_{1 \leq i < j \leq N} \text{Cov}(P^{(i)}, P^{(j)})$ is typically non-zero due to the simultaneous optimization of all components. This leads to an interdependence among components, where the model inadvertently captures interactions or spurious correlations between different components $P^{(i)}$ and $P^{(j)}$. As a result, the unified training process often leads to an inflated variance, increasing the risk of overfitting as the model attempts to capture not only the true signal but also the noise embedded in these interactions.

In contrast, the proposed *separable training* approach trains each component $P^{(i)}$ independently, such that:

$$\hat{P}_{out}^{(i)} = \arg\min_{\theta^{(i)}} \frac{1}{T} \sum_{t=1}^{T} (P^{(i)}(t) - \hat{P}_{out}^{(i)}(t; \theta^{(i)}))^2, \tag{14}$$

where $\hat{P}_{out}^{(i)}(t; \theta^{(i)})$ represents the model's prediction for component $P^{(i)}$ at time $t$ with parameter set $\theta^{(i)}$.

By optimizing each component $P^{(i)}$ separately, the covariance terms $\text{Cov}(P^{(i)}, P^{(j)})$ for $i \neq j$ are effectively nullified. Thus, under the separable training paradigm, the total variance simplifies to:

$$\sigma_{X_{In}}^2 = \sum_{i=1}^{N} \sigma_{P^{(i)}}^2, \tag{15}$$

since $\text{Cov}(P^{(i)}, P^{(j)}) = 0$ for all $i \neq j$ under separable training.

To further quantify the impact of separable training on variance reduction, we introduce the concept of a *variance reduction ratio $R$*, defined as the ratio of the total variance under separable training

$\sigma_{X_{In}}^2$ $^{\text{sep}}$ to that under unified training $\sigma_{X_{In}}^2$ $^{\text{uni}}$:

$$R = \frac{\sum_{i=1}^{N} \sigma_{P^{(i)}}^2}{\sum_{i=1}^{N} \sigma_{P^{(i)}}^2 + 2\sum_{1 \le i < j \le N} \text{Cov}(P^{(i)}, P^{(j)})}. \tag{16}$$

Since $\sum_{i<j} \text{Cov}(P^{(i)}, P^{(j)}) \ge 0$, it follows that $R \le 1$. This inequality directly implies that separable training results in a lower or, at worst, equivalent variance compared to unified training. The greater the sum of covariances $2\sum_{i<j} \text{Cov}(P^{(i)}, P^{(j)})$, the more significant the variance reduction achieved through separable training.

The elimination of these covariance terms in separable training not only reduces overall variance but also mitigates the model's tendency to overfit. This is because the absence of inter-component interactions prevents the model from capturing noise or spurious correlations that do not contribute to the true signal, thereby enhancing the generalization capability of the model.

Moreover, the preservation of independent optimization for each component ensures that each $P^{(i)}$ is learned to its mean value accurately without interference from other components, thereby maintaining the model's unbiased nature. Consequently, separable training effectively manages the bias-variance tradeoff by significantly reducing variance while maintaining low bias.

In statistical learning theory, this translates to a model that adheres more closely to the *oracle property*, where each component $P^{(i)}$ is estimated as if the others were known in advance. This property reinforces the idea that separable training provides a more efficient and theoretically grounded approach to handling complex time series data, thereby achieving an optimal balance between bias and variance. □

## C STATIONARITY TEST ON BENCHMARKED DATASETS

Table 4: Stationarity testing result on weather each variable using ADF and KPSS tests

| Variable | ADF | | | KPSS | | |
|---|---|---|---|---|---|---|
| | Statistic | p-value | Stationarity | Statistic | p-value | Stationarity |
| p (mbar) | -8.140 | 1.04e-12 | O | 0.9420 | 0.01 | X |
| T (degC) | -8.407 | 2.15e-13 | O | 8.1786 | 0.01 | X |
| Tpot (K) | -8.430 | 1.88e-13 | O | 8.1251 | 0.01 | X |
| Tdew (degC) | -6.505 | 1.14e-08 | O | 10.2832 | 0.01 | X |
| rh (%) | -17.053 | 8.04e-30 | O | 8.1633 | 0.01 | X |
| VPmax (mbar) | -9.398 | 6.31e-16 | O | 7.9154 | 0.01 | X |
| VPact (mbar) | -6.155 | 7.39e-08 | O | 9.3269 | 0.01 | X |
| VPdef (mbar) | -16.633 | 1.66e-29 | O | 5.0522 | 0.01 | X |
| sh (g/kg) | -6.177 | 6.58e-08 | O | 9.2933 | 0.01 | X |
| H2OC (mmol/mol) | -6.175 | 6.65e-08 | O | 9.3067 | 0.01 | X |
| rho (g/m$^3$) | -7.944 | 3.26e-12 | O | 7.5689 | 0.01 | X |
| wv (m/s) | -229.279 | 0.0 | O | 0.0207 | 0.1 | O |
| max. wv (m/s) | -22.467 | 0.0 | O | 3.2738 | 0.01 | X |
| wd (deg) | -18.273 | 2.32e-30 | O | 0.8335 | 0.01 | X |
| rain (mm) | -29.619 | 0.0 | O | 0.1776 | 0.1 | O |
| raining (s) | -21.262 | 0.0 | O | 0.1328 | 0.1 | O |
| SWDR (W/m$^2$) | -35.205 | 0.0 | O | 7.2054 | 0.01 | X |
| PAR (µmol/m$^2$/s) | -35.376 | 0.0 | O | 7.3854 | 0.01 | X |
| max. PAR (µmol/m$^2$/s) | -34.085 | 0.0 | O | 7.6197 | 0.01 | X |
| Tlog (degC) | -8.614 | 6.37e-14 | O | 8.3066 | 0.01 | X |
| OT | -25.113 | 0.0 | O | 0.2279 | 0.1 | O |

## D EXPERIMENTAL SETTINGS

### D.1 ENVIRONMENT SETTING

We conduct experiments on multivariate time series forecasting. All experiments were conducted in the same software and hardware environments. UBUNTU 18.04 LTS, PYTHON 3.8.0, NUMPY 1.22.3, SCIPY 1.10.1, MATPLOTLIB 3.6.2, PYTORCH 2.0.1, CUDA 11.4, NVIDIA Driver 470.182.03 i9 CPU, and NVIDIA RTX A5000.

Table 5: Stationarity testing result on exchange rate each variable using ADF and KPSS tests

| Variable | ADF | | | KPSS | | |
|---|---|---|---|---|---|---|
| | Statistic | p-value | Stationarity | Statistic | p-value | Stationarity |
| 0 | -1.665 | 0.4492 | X | 5.2917 | 0.01 | X |
| 1 | -2.150 | 0.2250 | X | 1.2468 | 0.01 | X |
| 2 | -1.353 | 0.6048 | X | 5.3135 | 0.01 | X |
| 3 | -1.587 | 0.4903 | X | 10.5355 | 0.01 | X |
| 4 | -2.869 | 0.0491 | O | 2.4475 | 0.01 | X |
| 5 | -2.120 | 0.2365 | X | 3.9282 | 0.01 | X |
| 6 | -1.748 | 0.4067 | X | 7.9789 | 0.01 | X |
| OT | -1.728 | 0.4166 | X | 7.0661 | 0.01 | X |

Table 6: Stationarity testing result on ETTh1 each variable using ADF and KPSS tests

| Variable | ADF | | | KPSS | | |
|---|---|---|---|---|---|---|
| | Statistic | p-value | Stationarity | Statistic | p-value | Stationarity |
| HUFL | -8.5505 | 9.25e-14 | O | 6.4781 | 0.01 | X |
| HULL | -5.1691 | 1.02e-05 | O | 2.2266 | 0.01 | X |
| MUFL | -8.6212 | 6.10e-14 | O | 9.0633 | 0.01 | X |
| MULL | -4.9641 | 2.61e-05 | O | 1.7190 | 0.01 | X |
| LUFL | -5.7969 | 4.73e-07 | O | 2.0147 | 0.01 | X |
| LULL | -4.7727 | 6.13e-05 | O | 1.1559 | 0.01 | X |
| OT | -3.4880 | 0.0083 | O | 9.4621 | 0.01 | X |

## D.2 DATASETS

1. Weather dataset consists of measurements for 21 weather indicators, such as temperature and humidity, collected every 10 minutes throughout the year 2020 (Wu et al., 2021).

2. Exchange dataset includes exchange rate data among 8 different countries (Lai et al., 2018).

3. ETT (Electricity Transformer Temperature) comprises four datasets: two with hourly granularity and two with 15-minute granularity, recorded between July 2016 and July 2018. Each dataset contains seven features related to oil and load conditions of transformers (Zhou et al., 2021).

4. Electricity dataset tracks the hourly electricity consumption of 321 clients from 2012 to 2014.

5. Traffic dataset represents the road occupancy rates, capturing hourly data recorded by sensors on the San Francisco freeways between 2015 and 2016.

6. ILL (Influenza-like Illness) dataset is provided by the Centers for Disease Control and Prevention (CDC) of the United States, covering the period from 2002 to 2021.

## D.3 BASELINES

We evaluated our model compared to the following state-of-the-art baselines:

1. DLinear (Zeng et al., 2023) (Decomposition Linear) paper investigates the effectiveness of Transformer-based models for long-term time series forecasting (LTSF). The authors challenge the dominance of Transformers by introducing a simple one-layer linear model, LTSF-Linear, which surprisingly outperforms Transformer-based LTSF models on nine real-life datasets. They argue that Transformers may not be ideal for LTSF because of the temporal information loss caused by the permutation-invariant self-attention mechanism. Their experiments demonstrate that LTSF-Linear, with its simple structure and trend-seasonality decomposition, achieves superior performance compared to complex Transformer models, suggesting that simpler models could be more suitable for certain time series forecasting tasks.

2. PatchTST (Nie et al., 2022) is a Transformer-based model designed for multivariate time series forecasting and self-supervised representation learning. It introduces two key components: patching, where time series data is segmented into subseries-level patches to serve as input tokens, enhancing local semantic information and reducing computation; and channel-independence, where each channel contains a univariate time series that shares

Table 7: Stationarity testing result on ETTh2 each variable using ADF and KPSS tests

| Variable | ADF | | | KPSS | | |
|---|---|---|---|---|---|---|
| | Statistic | p-value | Stationarity | Statistic | p-value | Stationarity |
| HUFL | -6.5264 | 1.01e-08 | O | 8.8581 | 0.01 | X |
| HULL | -4.5542 | 0.0002 | O | 12.7572 | 0.01 | X |
| MUFL | -4.0446 | 0.0012 | O | 1.2424 | 0.01 | X |
| MULL | -4.4530 | 0.0002 | O | 8.1632 | 0.01 | X |
| LUFL | -2.4355 | 0.1320 | X | 16.5272 | 0.01 | X |
| LULL | -3.3408 | 0.0132 | O | 1.0814 | 0.01 | X |
| OT | -3.5971 | 0.0058 | O | 1.8443 | 0.01 | X |

Table 8: Stationarity testing result on electricity each variable using ADF and KPSS tests (showing statistics for a sample of 20 out of the total 321 variables).

| Variable | ADF | | | KPSS | | |
|---|---|---|---|---|---|---|
| | Statistic | p-value | Stationarity | Statistic | p-value | Stationarity |
| 0 | -6.8575 | 1.63e-09 | O | 1.0495 | 0.01 | X |
| 1 | -7.0165 | 6.71e-10 | O | 1.7130 | 0.01 | X |
| 2 | -5.0386 | 1.86e-05 | O | 4.6124 | 0.01 | X |
| 3 | -6.1206 | 8.87e-08 | O | 15.8351 | 0.01 | X |
| 4 | -4.9192 | 3.20e-05 | O | 1.3654 | 0.01 | X |
| 5 | -4.6755 | 9.35e-05 | O | 1.4239 | 0.01 | X |
| 6 | -6.6138 | 6.28e-09 | O | 0.6762 | 0.0157 | X |
| 7 | -12.2476 | 9.70e-23 | O | 1.2862 | 0.01 | X |
| 8 | -10.6022 | 6.13e-19 | O | 1.3742 | 0.01 | X |
| 9 | -12.0190 | 3.05e-22 | O | 1.2859 | 0.01 | X |
| 10 | -5.0968 | 1.42e-05 | O | 1.0898 | 0.01 | X |
| 11 | -22.8691 | 0.0 | O | 8.1358 | 0.01 | X |
| 12 | -15.0668 | 8.84e-28 | O | 0.6601 | 0.0172 | X |
| 13 | -7.5981 | 2.43e-11 | O | 0.7042 | 0.0132 | X |
| 14 | -6.5456 | 9.11e-09 | O | 0.8886 | 0.01 | X |
| 15 | -11.2439 | 1.78e-20 | O | 0.8630 | 0.01 | X |
| 16 | -8.4163 | 2.04e-13 | O | 6.2602 | 0.01 | X |
| 17 | -7.3335 | 1.11e-10 | O | 8.7768 | 0.01 | X |
| 18 | -8.7300 | 3.21e-14 | O | 2.9707 | 0.01 | X |
| 19 | -14.8180 | 1.98e-27 | O | 3.8465 | 0.01 | X |

the same embedding and Transformer weights across all series. This approach allows PatchTST to handle longer look-back windows and capture essential temporal information, leading to significant improvements in long-term forecasting accuracy over other Transformer-based models, particularly when dealing with large datasets.

3. iTransformer (Liu et al., 2023) is a modified Transformer architecture designed specifically for time series forecasting. Unlike traditional Transformers that embed multiple time series variables as temporal tokens, iTransformer takes an inverted approach by embedding each time series as variate tokens. This method allows the attention mechanism to capture multivariate correlations more effectively, while a feed-forward network learns nonlinear representations of individual time series. By focusing on the relationships among variates, iTransformer achieves enhanced performance, generalization across different variates, and improved handling of longer lookback windows, making it highly effective for complex multivariate time series forecasting tasks.

4. TimeMixer (Wang et al., 2024) is a neural network architecture designed for long-term time series forecasting. It employs a hierarchical design that captures temporal dependencies at multiple scales, efficiently modeling both short-term and long-term dependencies in time series data. The architecture consists of multiple layers of specialized blocks, each handling different temporal scales, which allows the model to capture complex patterns across varying time intervals. By combining these blocks, TimeMixer achieves state-of-the-art performance on benchmark datasets, demonstrating its ability to handle diverse time series forecasting tasks effectively.

## D.4 HYPERPARAMETERS

1. $h$ : forecasting horizon length

2. $I$ : input sequence length

Table 9: Stationarity testing result on traffic each variable using ADF and KPSS tests (showing statistics for a sample of 20 out of the total 883 variables).

| Variable | ADF | | | KPSS | | |
|---|---|---|---|---|---|---|
| | Statistic | p-value | Stationarity | Statistic | p-value | Stationarity |
| 0 | -15.1612 | 6.59e-28 | O | 1.1652 | 0.01 | X |
| 1 | -15.8957 | 8.44e-29 | O | 0.2681 | 0.1 | O |
| 2 | -15.0168 | 1.04e-27 | O | 0.9598 | 0.01 | X |
| 3 | -19.1828 | 0.00 | O | 0.6013 | 0.0225 | X |
| 4 | -16.9776 | 9.06e-30 | O | 0.1101 | 0.1 | O |
| 5 | -18.1970 | 2.41e-30 | O | 1.1706 | 0.01 | X |
| 6 | -20.0680 | 0.00 | O | 1.8860 | 0.01 | X |
| 7 | -16.0007 | 6.52e-29 | O | 0.6163 | 0.0212 | X |
| 8 | -14.8895 | 1.57e-27 | O | 5.0434 | 0.01 | X |
| 9 | -15.2753 | 4.66e-28 | O | 1.1001 | 0.01 | X |
| 10 | -16.6232 | 1.69e-29 | O | 0.4720 | 0.0480 | X |
| 11 | -14.2461 | 1.51e-26 | O | 2.9375 | 0.01 | X |
| 12 | -16.9995 | 8.75e-30 | O | 1.6541 | 0.01 | X |
| 13 | -17.3764 | 5.10e-30 | O | 3.2661 | 0.01 | X |
| 14 | -17.3334 | 5.39e-30 | O | 2.8093 | 0.01 | X |
| 15 | -16.2227 | 3.88e-29 | O | 2.7122 | 0.01 | X |
| 16 | -10.2786 | 3.83e-18 | O | 3.0002 | 0.01 | X |
| 17 | -13.0770 | 1.90e-24 | O | 3.4275 | 0.01 | X |
| 18 | -15.6792 | 1.48e-28 | O | 2.6290 | 0.01 | X |
| 19 | -13.6018 | 1.95e-25 | O | 1.4351 | 0.01 | X |

Table 10: Stationarity testing result on national illness(ILL) each variable using ADF and KPSS tests

| Variable | ADF | | | KPSS | | |
|---|---|---|---|---|---|---|
| | Statistic | p-value | Stationarity | Statistic | p-value | Stationarity |
| % WEIGHTED | -7.846 | 5.75e-12 | O | 0.1857 | 0.1 | O |
| % UNWEIGHTED | -7.747 | 1.03e-11 | O | 0.3008 | 0.1 | O |
| AGE 0-4 | -6.507 | 1.12e-08 | O | 1.6179 | 0.01 | X |
| AGE 5-24 | -6.383 | 2.20e-08 | O | 1.2751 | 0.01 | X |
| ILITOTAL | -6.161 | 7.16e-08 | O | 1.7705 | 0.01 | X |
| OF PROVIDERS | -1.713 | 0.4243 | X | 3.6161 | 0.01 | X |
| OT | -0.982 | 0.7598 | X | 4.0449 | 0.01 | X |

3. $K$ : kernel size of time-series decomposition method

4. $p$ : period length of time-series decomposition method

5. $\lambda$ : learning rate

6. decomp.method : decomposition method

# E   COMPUTATIONAL TIME AND MODEL USAGE

This section compares the time per epoch and memory usage between the original model and the proposed IDEAS framework during the training process (cf. Table 15 and Table E). The proposed IDEAS applies separable training, where multiple stages of training are conducted. For instance, if the model undergoes 3 iterations, it learns 3 distinct predictable patterns separately. In this process, the time taken per epoch is similar to or shorter than that of the original model because the training data is processed in a more compact form than the raw data. Furthermore, since each predictable pattern becomes simpler for the model to learn, fewer epochs are required for convergence at each stage. For example, if a model originally requires a total of 50 epochs for training, in separable training, each stage may converge within 10 to 15 epochs. Thus, even with training across 3 stages, only about 45 epochs in total are required, leading to a reduction in overall computation time.

# F   ALGORITHM OF IDEAS

In this section, we provide the IDEAS algorithm along with the overall architecture diagram to help you understand the detailed algorithm of the proposed method.

Table 11: DLinear (IDEAS) Hyperparameters used for each dataset.

| Datasets | Weather | | | | Exchange | | | | ETTh1 | | | |
|---|---|---|---|---|---|---|---|---|---|---|---|---|
| $h$ | 96 | 192 | 336 | 720 | 96 | 192 | 336 | 720 | 96 | 192 | 336 | 720 |
| $I$ | 336 | 336 | 336 | 480 | 336 | 336 | 336 | 336 | 336 | 336 | 192 | 336 |
| $k$ | 19 | 48 | 54 | 60 | 60 | 60 | 60 | 60 | 13 | 25 | 48 | 29 |
| $p$ | 12 | 6 | 6 | 36 | 7 | 7 | 7 | 7 | 12 | 12 | 24 | 12 |
| $\lambda$ | 0.001 | 0.001 | 0.001 | 0.00125 | 0.005 | 0.005 | 0.005 | 0.005 | 0.005 | 0.000125 | 0.000125 | 0.000125 |
| Decomp. method | STR | STR | STR | STR | STR | STR | STR | STR | STL | STL | STR | STL |
| Datasets | ETTm1 | | | | ETTh2 | | | | ETTm2 | | | |
| $h$ | 96 | 192 | 336 | 720 | 96 | 192 | 336 | 720 | 96 | 192 | 336 | 720 |
| I | 336 | 336 | 336 | 336 | 336 | 336 | 336 | 336 | 336 | 336 | 336 | 336 |
| $k$ | 24 | 12 | 12 | 12 | 42 | 36 | 42 | 42 | 48 | 48 | 48 | 48 |
| $p$ | 48 | 24 | 24 | 24 | 12 | 24 | 12 | 12 | 48 | 48 | 48 | 48 |
| $\lambda$ | 0.0025 | 0.00025 | 0.00025 | 0.0025 | 0.005 | 0.0025 | 0.005 | 0.005 | 0.0025 | 0.0025 | 0.0025 | 0.0025 |
| Decomp. method | STR | STR | STR | STR | STL | STR | STL | STL | STR | STR | STR | STR |
| Datasets | Electricity | | | | Traffic | | | | ILL | | | |
| $h$ | 96 | 192 | 336 | 720 | 96 | 192 | 336 | 720 | 24 | 36 | 48 | 60 |
| I | 336 | 336 | 336 | 336 | 336 | 720 | 336 | 336 | 104 | 104 | 104 | 104 |
| $k$ | 25 | 25 | 49 | 49 | 7 | 3 | 7 | 5 | 7 | 7 | 7 | 7 |
| $p$ | 12 | 12 | 12 | 12 | 12 | 12 | 12 | 12 | 12 | 12 | 12 | 12 |
| $\lambda$ | 0.0025 | 0.001 | 0.005 | 0.001 | 0.001 | 0.00025 | 0.001 | 0.025 | 0.005 | 0.005 | 0.005 | 0.005 |
| Decomp. method | STL | STL | STL | STL | STL | STR | STL | STL | STL | STL | STL | STL |

Table 12: PatchTST (IDEAS) Hyperparameters used for each dataset

| Datasets | Weather | | | | Exchange | | | | ETTh1 | | | |
|---|---|---|---|---|---|---|---|---|---|---|---|---|
| $h$ | 96 | 192 | 336 | 720 | 96 | 192 | 336 | 720 | 96 | 192 | 336 | 720 |
| $I$ | 336 | 336 | 336 | 720 | 336 | 336 | 336 | 336 | 336 | 336 | 336 | 336 |
| $k$ | 35 | 42 | 42 | 42 | 36 | 30 | 6 | 12 | 25 | 25 | 25 | 24 |
| $p$ | 42 | 14 | 35 | 28 | 24 | 6 | 7 | 6 | 12 | 12 | 12 | 24 |
| $\lambda$ | 0.0001 | 0.00001 | 0.0005 | 0.0005 | 0.00001 | 0.00025 | 0.000025 | 0.00025 | 0.000025 | 0.000025 | 0.000025 | 0.00001 |
| Decomp. method | STR | STR | STR | STR | STR | STR | STR | STR | STL | STL | STL | STL |
| Datasets | ETTm1 | | | | ETTh2 | | | | ETTm2 | | | |
| $h$ | 96 | 192 | 336 | 720 | 96 | 192 | 336 | 720 | 96 | 192 | 336 | 720 |
| I | 336 | 336 | 336 | 336 | 336 | 336 | 336 | 336 | 336 | 336 | 336 | 336 |
| $k$ | 7 | 7 | 7 | 7 | 42 | 9 | 23 | 19 | 11 | 11 | 11 | 11 |
| $p$ | 12 | 12 | 12 | 12 | 24 | 12 | 12 | 12 | 12 | 12 | 12 | 12 |
| $\lambda$ | 0.001 | 0.001 | 0.001 | 0.001 | 0.000125 | 0.0001 | 0.00025 | 0.001 | 0.0025 | 0.0025 | 0.0025 | 0.0025 |
| Decomp. method | STR | STR | STR | STR | STR | STL | STL | STL | STR | STR | STR | STR |
| Datasets | Electricity | | | | Traffic | | | | ILL | | | |
| $h$ | 96 | 192 | 336 | 720 | 96 | 192 | 336 | 720 | 24 | 36 | 48 | 60 |
| I | 336 | 336 | 336 | 336 | 336 | 336 | 336 | 336 | 104 | 104 | 104 | 104 |
| $k$ | 13 | 13 | 13 | 13 | 7 | 7 | 7 | 7 | 9 | 9 | 7 | 29 |
| $p$ | 12 | 12 | 12 | 12 | 12 | 12 | 12 | 12 | 12 | 12 | 12 | 12 |
| $\lambda$ | 0.0025 | 0.0025 | 0.0025 | 0.001 | 0.001 | 0.001 | 0.001 | 0.005 | 0.025 | 0.025 | 0.025 | 0.025 |
| Decomp. method | STL | STL | STL | STL | STL | STL | STL | STL | STL | STL | STL | STL |

Table 13: iTransformer (IDEAS) Hyperparameters used for each dataset

| Datasets | Weather | | | | Exchange | | | | ETTh1 | | | |
|---|---|---|---|---|---|---|---|---|---|---|---|---|
| $h$ | 96 | 192 | 336 | 720 | 96 | 192 | 336 | 720 | 96 | 192 | 336 | 720 |
| $I$ | 192 | 192 | 192 | 336 | 336 | 336 | 336 | 336 | 336 | 336 | 192 | 336 |
| $k$ | 48 | 48 | 48 | 48 | 60 | 60 | 60 | 60 | 9 | 23 | 24 | 24 |
| $p$ | 6 | 6 | 6 | 6 | 7 | 7 | 28 | 7 | 12 | 12 | 24 | 24 |
| $\lambda$ | 0.0005 | 0.005 | 0.005 | 0.005 | 0.000125 | 0.00025 | 0.001 | 0.001 | 0.000025 | 0.000125 | 0.000125 | 0.000125 |
| Decomp. method | STR | STR | STR | — | STR | STR | STR | STR | STL | STL | STR | STR |
| Datasets | ETTm1 | | | | ETTh2 | | | | ETTm2 | | | |
| $h$ | 96 | 192 | 336 | 720 | 96 | 192 | 336 | 720 | 96 | 192 | 336 | 720 |
| $I$ | 336 | 336 | 336 | 336 | 336 | 336 | 336 | 336 | 336 | 336 | 336 | 336 |
| $k$ | 30 | 30 | 30 | 30 | 42 | 42 | 42 | 42 | 24 | 24 | 24 | 24 |
| $p$ | 24 | 24 | 24 | 24 | 24 | 24 | 24 | 24 | 24 | 24 | 24 | 24 |
| $\lambda$ | 0.0025 | 0.001 | 0.0025 | 0.001 | 0.000025 | 0.000025 | 0.000025 | 0.001 | 0.001 | 0.001 | 0.001 | — |
| Decomp. method | STR | STR | STR | STR | STR | STR | STR | STR | STR | STR | STR | STR |
| Datasets | Electricity | | | | Traffic | | | | ILL | | | |
| $h$ | 96 | 192 | 336 | 720 | 96 | 192 | 336 | 720 | 24 | 36 | 48 | 60 |
| I | 336 | 336 | 336 | 336 | 336 | 336 | 336 | 336 | 144 | 104 | 104 | 104 |
| $k$ | 7 | 7 | 7 | 7 | 9 | 9 | 9 | 9 | 7 | 9 | 7 | 7 |
| $p$ | 24 | 24 | 24 | 24 | 12 | 12 | 12 | 12 | 12 | 12 | 12 | 12 |
| $\lambda$ | 0.0001 | 0.0001 | 0.0001 | 0.0001 | 0.005 | 0.005 | 0.005 | 0.005 | 0.005 | 0.000025 | 0.000025 | 0.000025 |
| Decomp. method | STR | STR | STR | STR | STL | STL | STL | STL | STL | STL | STL | STL |

Table 14: TimeMixer (IDEAS) Hyperparameters used for each dataset

| Datasets | Weather | | | | Exchange | | | | ETTh1 | | | |
|---|---|---|---|---|---|---|---|---|---|---|---|---|
| $h$ | 96 | 192 | 336 | 720 | 96 | 192 | 336 | 720 | 96 | 192 | 336 | 720 |
| $I$ | 96 | 96 | 192 | 96 | 336 | 96 | 96 | 96 | 336 | 336 | 336 | 336 |
| $k$ | 14 | 14 | 14 | 14 | 7 | 12 | 12 | 12 | 9 | 9 | 43 | 47 |
| $p$ | 12 | 12 | 12 | 12 | 12 | 8 | 8 | 8 | 12 | 12 | 12 | 12 |
| $\lambda$ | 0.01 | 0.01 | 0.01 | 0.01 | 0.01 | 0.01 | 0.01 | 0.01 | 0.001 | 0.001 | 0.005 | 0.005 |
| Decomp. method | STR | STR | STR | STR | STL | STR | STR | STR | STL | STL | STL | STL |
| Datasets | ETTm1 | | | | ETTh2 | | | | ETTm2 | | | |
| $h$ | 96 | 192 | 336 | 720 | 96 | 192 | 336 | 720 | 96 | 192 | 336 | 720 |
| $I$ | 336 | 336 | 336 | 336 | 336 | 336 | 336 | 336 | 336 | 336 | 336 | 336 |
| $k$ | 24 | 24 | 24 | 24 | 48 | 9 | 23 | 9 | 47 | 47 | 47 | 47 |
| $p$ | 12 | 12 | 12 | 12 | 12 | 12 | 12 | 12 | 12 | 12 | 12 | 12 |
| $\lambda$ | 0.01 | 0.01 | 0.001 | 0.001 | 0.01 | 0.005 | 0.001 | 0.01 | 0.000125 | 0.001 | 0.001 | 0.001 |
| Decomp. method | STR | STR | STR | STR | STR | STL | STL | STL | STR | STR | STR | STR |
| Datasets | Electricity | | | | Traffic | | | | ILL | | | |
| $h$ | 96 | 192 | 336 | 720 | 96 | 192 | 336 | 720 | 24 | 36 | 48 | 60 |
| I | 336 | 336 | 336 | 336 | 336 | 336 | 336 | 336 | 144 | 144 | 104 | 104 |
| $k$ | 7 | 7 | 7 | 7 | 7 | 7 | 7 | 7 | 43 | 23 | 9 | 23 |
| $p$ | 12 | 12 | 12 | 12 | 12 | 12 | 12 | 12 | 24 | 12 | 12 | 12 |
| $\lambda$ | 0.01 | 0.01 | 0.01 | 0.01 | 0.01 | 0.01 | 0.01 | 0.01 | 0.005 | 0.01 | 0.001 | 0.01 |
| Decomp. method | STL | STL | STL | STL | STL | STL | STL | STL | STL | STL | STL | STL |

Table 15: Computational time per 1 epoch

| Models | Weather | Exchange | ETTh1 | ETTm1 | ETTh2 | ETTm2 | Electricity | Traffic | ILL |
|---|---|---|---|---|---|---|---|---|---|
| DLinear | 7.824 | 3.699 | 1.898 | 6.371 | 2.011 | 4.324 | 9.964 | 13.78 | 1.186 |
| DLinear (IDEAS) | 10.70 | 3.458 | 3.339 | 4.383 | 3.518 | 8.492 | 14.37 | 12.40 | 0.956 |
| PatchTST | 79.57 | 5.297 | 21.31 | 10.81 | 22.95 | 4.512 | 208.6 | 312.8 | 3.574 |
| PatchTST (IDEAS) | 75.42 | 5.391 | 3.881 | 11.27 | 2.967 | 3.812 | 218.9 | 316.7 | 2.002 |
| iTransformer | 21.30 | 7.254 | 16.84 | 30.28 | 6.782 | 35.91 | 42.88 | 89.84 | 6.884 |
| iTransformer (IDEAS) | 21.45 | 6.687 | 14.49 | 27.39 | 5.294 | 36.84 | 35.56 | 117.9 | 7.217 |
| TimeMixer | 48.51 | 11.90 | 31.88 | 201.8 | 49.21 | 72.81 | 314.9 | 901.5 | 5.073 |
| TimeMixer (IDEAS) | 40.65 | 5.179 | 15.27 | 76.11 | 16.20 | 25.31 | 213.7 | 673.1 | 1.544 |

Table 16: Model usage (MB)

| Models | Weather | Exchange | ETTh1 | ETTm1 | ETTh2 | ETTm2 | Electricity | Traffic | ILL |
|---|---|---|---|---|---|---|---|---|---|
| DLinear | 21.87 | 19.11 | 23.35 | 19.09 | 23.35 | 23.35 | 67.61 | 148.8 | 17.89 |
| DLinear (IDEAS) | 21.87 | 19.11 | 20.90 | 21.00 | 23.35 | 23.35 | 67.61 | 116.3 | 17.89 |
| PatchTST | 4,102 | 398.5 | 571.5 | 84.57 | 370.0 | 452.8 | 1,543 | 1,628 | 170.1 |
| PatchTST (IDEAS) | 4,071 | 401.4 | 569.0 | 80.82 | 367.4 | 468.5 | 1,058 | 1,540 | 63.05 |
| iTransformer | 35.53 | 98.21 | 121.8 | 123.7 | 128.7 | 122.9 | 601.8 | 3,821 | 80.21 |
| iTransformer (IDEAS) | 253.2 | 163.2 | 198.1 | 159.7 | 158.8 | 159.7 | 866.6 | 5,336 | 186.8 |
| TimeMixer | 2,011 | 902.4 | 5,803 | 178.5 | 101.8 | 614.9 | 980.4 | 6,109 | 140.8 |
| TimeMixer (IDEAS) | 2,338 | 1,129 | 7,483 | 253.4 | 127.3 | 828.2 | 1,268 | 8,546 | 165.0 |

# G    COMPARISON WITH EXISTING APPROACHES

In this section, we provide a comprehensive comparison between the IDEAS framework and existing approaches that share some similarities in concept but differ in execution and objective. This comparison aims to highlight the unique aspects of the IDEAS methodology, particularly the combination of iterative residual decomposition and Separable Training.

**Overview of existing residual-based methods:**    Residual-based methods, such as Boosting Algorithms (e.g., Gradient Boosting, AdaBoost), iteratively learn from residuals by refining predictions based on the errors of the previous models. These methods share a conceptual similarity with our iterative residual decomposition approach, as both aim to minimize residual errors over successive iterations. However, unlike iterative residual decomposition, which iteratively decomposes time series data to extract meaningful patterns, boosting algorithms primarily focus on reducing residuals

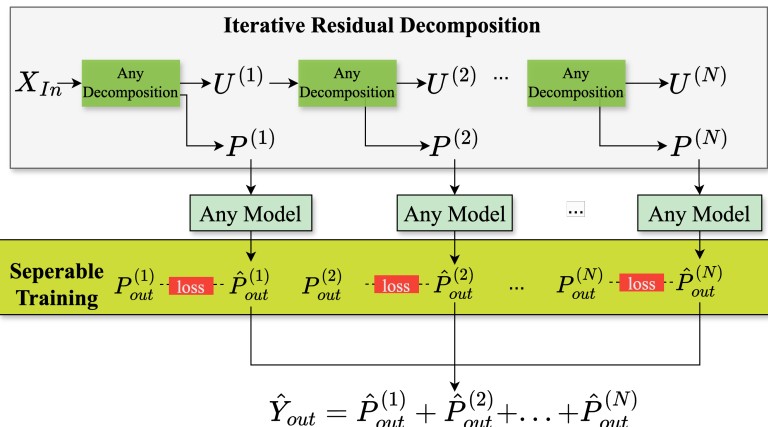

---

**Algorithm 2:** IDEAS

---

**Input:** Input time series data $X_{In}$, Any forecasting model $\theta_f^{(i)}$ for each component $i$, MSE loss function $L$, maximum residual iteration number $max\_iter = N$, maximum train iteration number $train\_iter = M$

1   $i \leftarrow 1$;

2   Decompose $X_{In}$ into $P^{(1)}$ and $U^{(1)}$;

3   Initialize model parameters $\theta_f^{(1)}$;

4   **while** $i \leq max\_iter$ **do**

5      Train model $\theta_f^{(i)}$ on component $P^{(i)}$ using:

$$\hat{P}_{out}^{(i)} = \theta_f^{(i)}(P^{(i)})$$

       **while** $j \leq train\_iter$ **do**

6          Update model parameters $\theta_f^{(i)}$ using gradient descent with loss:

$$\theta_f^{(i)} := \theta_f^{(i)} - \eta \nabla_{\theta_f^{(i)}} L(P_{out}^{(i)}, \hat{P}_{out}^{(i)})$$

         $j \leftarrow j + 1$;

7      Decompose $U^{(i)}$ into $P^{(i+1)}$ and $U^{(i+1)}$;

8      Initialize model parameters $\theta_f^{(i+1)}$;

9      $i \leftarrow i + 1$;

10   **return** final prediction $\hat{Y}_{out} = \sum_{i=1}^{N} \hat{P}_{out}^{(i)}$

---

within machine learning models and do not perform decomposition iteratively on the time series itself.

**Separable training and ensemble learning approaches:** Ensemble learning methods, such as Bagging and Stacking, train different models on various subsets or aspects of the data and then aggregate their predictions. While this shares the notion of training models separately, it fundamentally differs from IDEAS's separable training, which explicitly decomposes and trains individual time series components (e.g., trend, seasonality, residuals) independently. This ensures that each component is learned in a focused manner, unlike ensemble methods that do not distinguish between the underlying structures of the data.

**Recent time series decomposition models:** Recent models like DLinear and Autoformer have incorporated decomposition techniques to better handle the inherent complexity of time series data. These models often decompose time series into different components for better prediction accuracy, similar to our decomposition process. However, unlike the iterative nature of the iterative residual decomposition in IDEAS, where decomposition is repeatedly applied to refine residuals until they converge towards white noise, these models typically apply decomposition only once. This iterative

refinement sets IDEAS apart in its ability to extract and reduce overlooked predictable patterns more effectively.

**Hybrid approaches in time series forecasting:** Hybrid models combining ARIMA with deep learning models aim to leverage the strengths of both traditional statistical methods and modern machine learning techniques for time series forecasting. While they handle different aspects of the data, they do not achieve the systematic separation and independent training of components as in IDEAS's separable training. The iterative residual decomposition and Separable Training combination provides a more structured way of addressing noise and patterns than simply combining model outputs.

In summary, while there exist approaches that share certain aspects with IDEAS, such as residual learning, ensemble learning, and decomposition-based methods, none of them achieve the same level of effectiveness in addressing the bias-variance tradeoff in time series forecasting. The iterative residual decomposition provides an unbiased, systematic extraction of predictable patterns, while separable training ensures that each component is learned without interference from others, leading to improved generalization and forecasting accuracy. This combination is unique to IDEAS and represents a significant advancement in handling the complexity and non-stationarity inherent in real-world time series data.

## H VISUALIZATION

### H.1 RESIDUAL DISTRIBUTIONS

In Figure 7, we visualize the residual distributions on 7 datasets.

### H.2 DURBIN-WATSON AND ACF

In Figure 8, we visualize the statistical measures(Durbin-Watson and ACF) on 6 datasets.

To complement this theoretical analysis, we empirically validate Theorem 1 by conducting ACF and Durbin-Watson tests across multiple iterations of the iterative residual decomposition. As shown in Figure 8, the Durbin-Watson statistic (blue line) approaches the ideal value of $2.0$ (red dashed line), and the ACF statistic (orange line) converges towards $0.0$ (pink dashed line) as the number of iterations increases. This convergence indicates that the residuals become progressively closer to white noise, supporting our theorem.

### H.3 SENSITIVITY ON SEPARABLE TRAINING ITERATIONS

In Figure 9, we show the sensitivity to iteration number $N$ of separable training on 6 datasets.

### H.4 VISUALIZAITON OF PREDICTABLE PATTERNS

In Figure 10, we visualize the predictable patterns on 8 datasets.

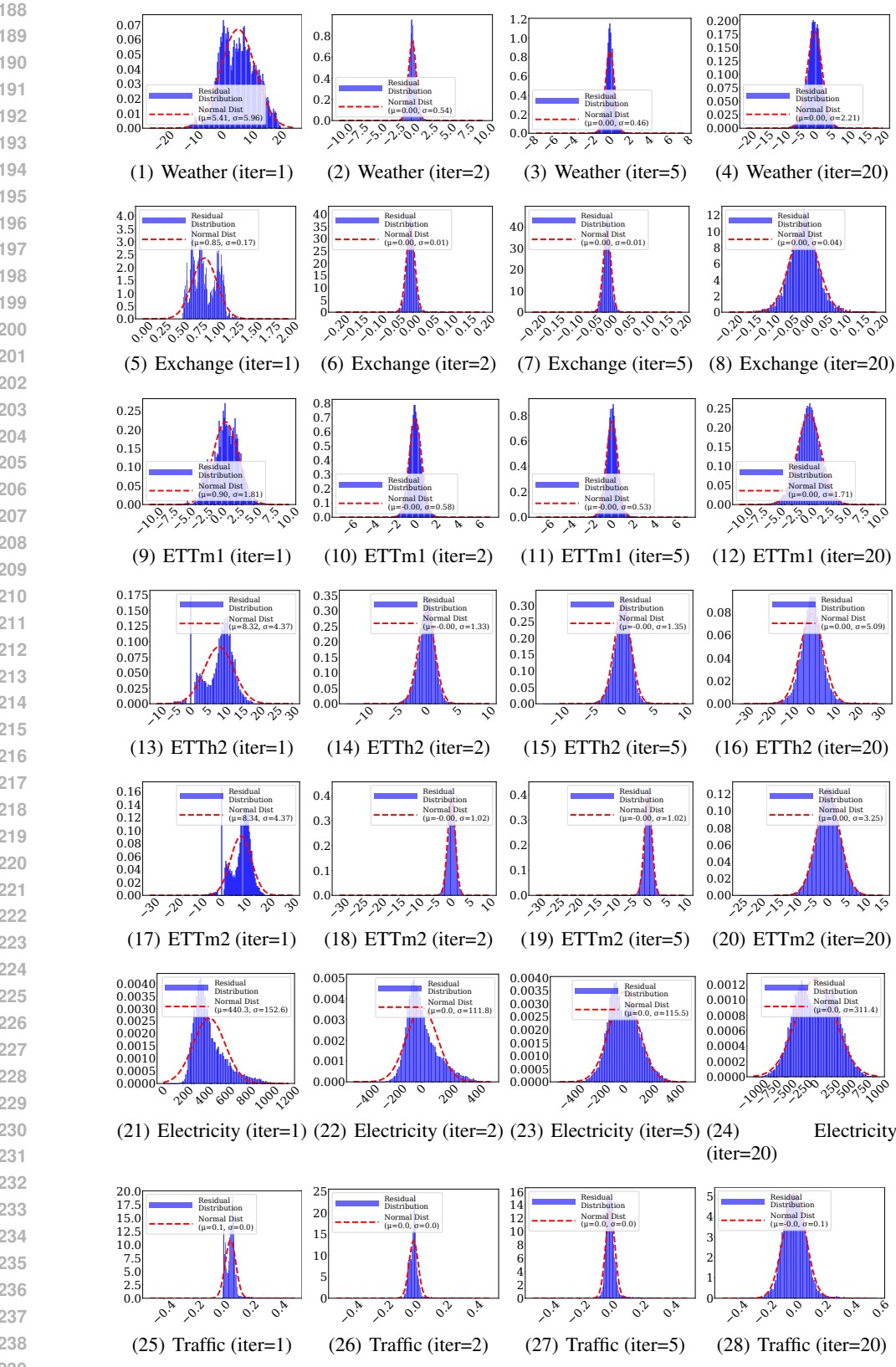

Figure 7: Visualization of the residual distributions over successive iterations for the 7 datasets. As the iteration process progresses, the residuals (blue bars) increasingly resemble a normal distribution (red dashed line), indicating that STL decomposition method iteratively refines the residuals towards a more Gaussian-like distribution.

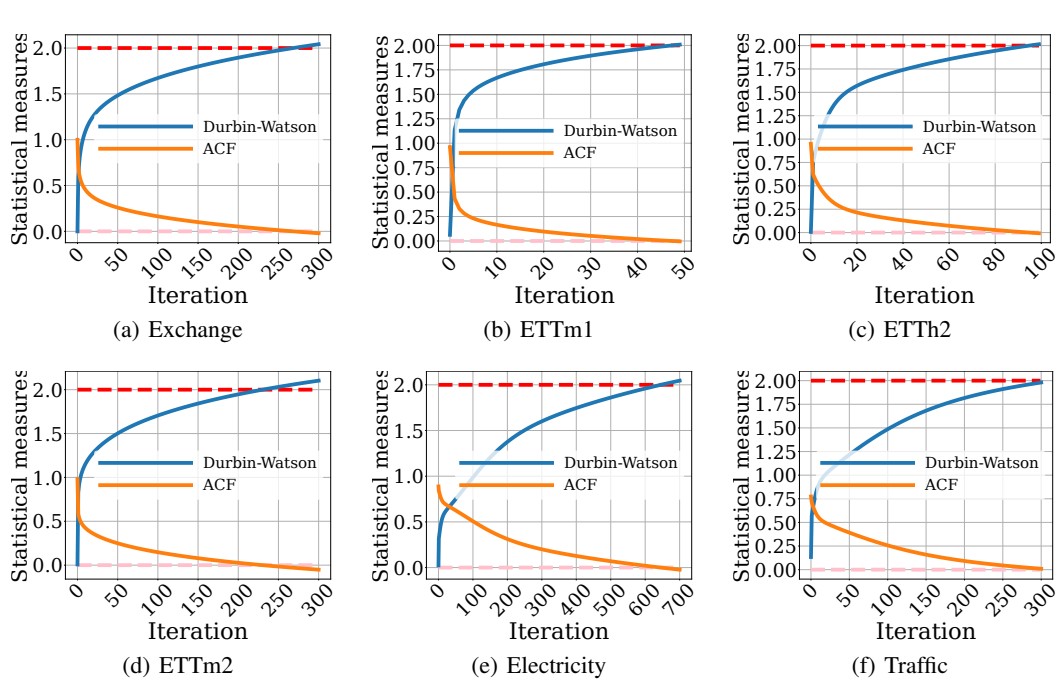

Figure 8: Empirical validation showing that as iterations increase, the residuals approach white noise, demonstrated by the Durbin-Watson and ACF statistics converging towards their expectation.

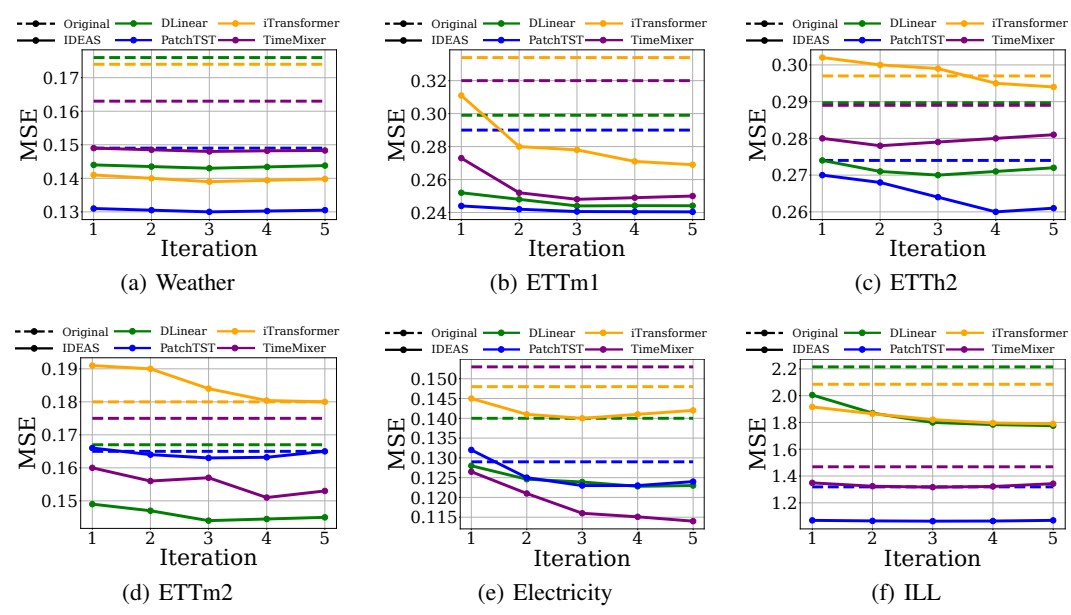

Figure 9: Sensitivity to iteration number $N$ of separable training.

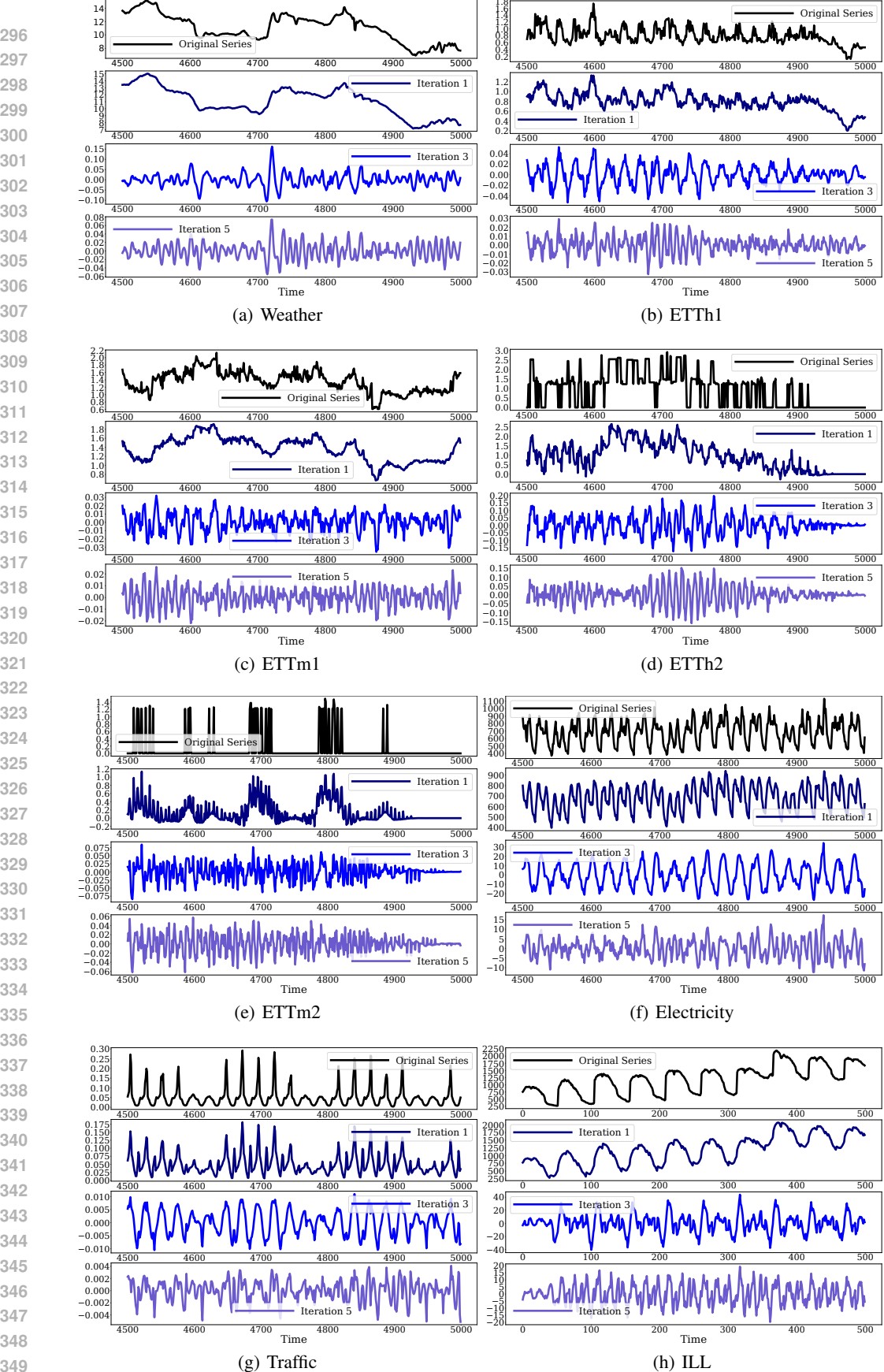

Figure 10: Visualization of the original time series (top) and the predictable patterns at each iteration of the iterative residual decomposition.

