# OpenReview forum: "Enhancing Time-Series Forecasting with Iterative Decomposition and Separable Training"
_ICLR.cc/2025/Conference — ICLR 2025 Conference Withdrawn Submission_

### Official Review · Reviewer_niXv · 2024-10-30

**Soundness:** 3
**Presentation:** 3
**Contribution:** 1
**Rating:** 5
**Confidence:** 4

**Summary:**

This paper introduces a time series training framework based on an iterative time series decomposition architecture. The proposed method, called IDEAS, consists of two parts. The first part iteratively separates predictable patterns from unpredictable noise. The second part, called separable training, is used to effectively manage the bias-variance trade-off and balance between model complexity and generalization. The superiority of the IDEAS framework over conventional time series decomposition techniques is thoroughly illustrated in experiments conducted on 9 datasets and 4 state-of-the-art models.

**Strengths:**

* Multiple statistical tests are utilized to ensure that the residuals of the iterative decomposition method eventually reach a normal distribution, indicating that all predictable patterns have been extracted and only noise remains.
* The iterative residual decomposition method is both empirically evaluated on real-world datasets and theoretically proven to achieve unbiasedness.
* A reasonable number of datasets and state-of-the-art forecasting models are used in the experiments to demonstrate the superiority of the proposed decomposition algorithm over conventional time series decomposition methods.
* Table 15 illustrates results in terms of computational complexity, which further strengthens the experiments.

**Weaknesses:**

* The memory usage results in Table 16 of the Appendix are somewhat concerning. For example, the memory usage for iTransformer on the Weather dataset increases by about 900% with IDEAS, which could be problematic under limited computational resources.
* The proposed method is not an entirely novel decomposition technique. The first part involves iterative usage of conventional time series decomposition methods, and the second part involves training and aggregating the predictions of each extracted pattern from the dataset. The novelty appears limited.

**Questions:**

* The approach in this work seems to involve iteratively utilizing widely used conventional decomposition techniques. Has this idea, which is inherently similar to the concept of boosting, never been explored in the literature? If there are similar works, could the authors provide an additional related work subsection on this topic?
* Can you provide more information on the feature selection process under IDEAS? Are the same features extracted for each pattern? Are there any features specifically related to the iterative decomposition process?

---

### Official Review · Reviewer_aqho · 2024-11-01

**Soundness:** 2
**Presentation:** 1
**Contribution:** 2
**Rating:** 3
**Confidence:** 4

**Summary:**

The paper proposes iterative use of STL or STR decomposition of time-series to separate out a series of predictable components from the time-series. Then each of those components are forecasted using some forecasting algorithm and the forecasts are added up to generate the final forecasts. The accuracy is then compared to the original forecasting algorithm directly applied on the original time-series. Results are shown on the usual long-horizon datasets that were used since the [informer](https://arxiv.org/abs/2012.07436) paper.

**Strengths:**

1. The method is simple and generally applicable to any forecasting model or algorithm.

2. It seems to show some benefit uniformly across the informer datasets.

3. It is verified with 4 deep learning forecasting models, showing some robustness to the idea.

**Weaknesses:**

1. The theory in the paper actually does not show anything and in my opinion can be omitted.

2. I think now there are much more extensive benchmarks that can be used to show the true robustness of the method.

3. Most of the heavy lifting is done by the STL method in question.

4. Writing is unclear and sometimes confusing/wrong in places. The paper comes up with some terms which are not clarified in teh main text and sometimes not even in the references.

5. No comparison to residual MLP models like N-BEATS, N-HITS and TiDE.

**Questions:**

1. Writing can be improved significantly. Please define terms like variable-centered learning. Also please define the specific STL and STR methods used and the difference between the two for completeness. Bias-variance trade-off is common to even usual regression and classification problems and the time-series specific differences are not clear in the writing. In line 120, why do transformer models reduce overfitting? In general this whole section is a bit vacuous.

Line 132-134 discusses problems with STL, STR methods while this paper applies the same iteratively. Is the hope that iterative application of the same methods gets rid of these limitations. That is not clear to me.

2. I think both the theorems don't add any value in fact they are kind of trivial or wrong. The first theorem basically assumes the result of the theorem it self. I don't see what is the point of that.

The second theorem firstly assumes that the components $P^(i)$'s are correlated and therefore the covariance cross terms appear in the variance expression. This actually makes separating them the core problem which is offloaded to STL algorithms. So what is the contribution of this theorem exactly? We all know that cross terms appear in variance calculations.

3.  I think the benchmarks should be extended to include Monash datasets like in [chronos](https://arxiv.org/abs/2403.07815), [moirai](https://arxiv.org/abs/2402.02592), [timesfm](https://arxiv.org/pdf/2310.10688). I think these benchmarks contain more diverse and sometimes less stationary time-series.

4. MLP models like N-HiTS, N-BEATS separate out residual components in their MLP blocks. I think we should show whether this methods are still relevant for such models or not.

---

### Official Review · Reviewer_ZXqu · 2024-11-02

**Soundness:** 2
**Presentation:** 3
**Contribution:** 2
**Rating:** 5
**Confidence:** 3

**Summary:**

## Summary
This paper introduces a framework called IDEAS to enhance time series forecasting. IDEAS aims to address the bias-variance trade-off issue by employing Iterative Residual Decomposition and Separable Training, effectively separating predictable patterns from unpredictable noise within time series data. The framework is applied to existing time series decomposition and forecasting models. While IDEAS shows performance gains on nine benchmark datasets across four state-of-the-art models, the results are mixed, with some datasets demonstrating notable improvements and others not.

**Strengths:**

## Strengths
1.	**Innovation**: The IDEAS framework offers a novel approach using decomposition for tackling the bias-variance trade-off in time series forecasting task.

2.	**Experiment**: The paper tests the framework on multiple benchmark datasets and models, showcasing its potential generalizability.

**Weaknesses:**

## Weaknesses
1.	**Mixed Experimental Results**: While some datasets show significant improvements, others do not, suggesting the framework's effectiveness may vary depending on dataset characteristics.
2.	**Insufficient Discussion of Decomposition Algorithm**: The decomposition algorithm is introduced, but the number of iterations of decomposition is not discussed.
3.	**Lack of Novelty**: The idea of combining the existing decomposition method with the existing time series forecasting method weakens the innovation of this method.

**Questions:**

## Clarification Questions
1.	**Theoretical Details**: Could the paper provide more rigorous theoretical proofs to strengthen its foundation of forecasting results?
2.	**Decomposition Algorithm**: How is the number of decomposition algorithm iterations chosen?
3.	**Dataset Analysis**: For the datasets where IDEAS performs well versus those where it does not, could the paper clarify characteristics that contribute to these differences?

---

### Official Review · Reviewer_HKjH · 2024-11-04

**Soundness:** 1
**Presentation:** 2
**Contribution:** 2
**Rating:** 3
**Confidence:** 4

**Summary:**

The paper proposes a method to improve time series forecasting (TSF) by iteratively decomposing the time series data into multiple predictable components and then training a TSF model separately for each component. This approach aims to address overfitting and underfitting by isolating different characteristics of the data and optimizing model training on each separately.

**Strengths:**

- The idea is compelling. Decomposition yields simpler time-series (e.g., single-periodic), which are easier for TSF models to learn.
- The proposed method is a meta-framework that integrates seamlessly with existing forecasting models.
- Evaluations using multiple base models demonstrate significant performance improvements.

**Weaknesses:**

- Invalid theoretical proofs: Theoretical justifications provided for the IDEAS framework are highly questionable. Theorem 1's "proof" merely restates the theorem without proper derivation, and Theorem 2 relies on key mathematical assumptions that are poorly justified, such as the interpretation of covariance and variance. For instance, the authors seem unaware that covariance can be negative and that data variance does not equate to model variance, leading to unclear interpretations of variance reduction.

- Lack of clarity in methodology: The paper does not specify the exact decomposition algorithm(s) employed in the experiments, making it difficult to understand how IDEAS might perform with alternative decomposition techniques. If the method employed is STL, it raises the question of whether the framework is suitable only for multi-periodic time-series data or if it can be generalized to other types.

**Questions:**

- Which specific decomposition algorithms were employed in the experiments, and how could the choice of decomposition algorithms influence the performance of IDEAS?
- Is the performance improvement attributable to the increased number of model parameters? If the exact same model architecture is used, IDEAS utilizes significantly more parameters than the baseline, potentially making the comparison unfair.

---

### Note · Authors · 2024-12-05

**Comment:**

Thank you for all the sincere reviews, I would like to withdraw my paper.

**Withdrawal Confirmation:**

I have read and agree with the venue's withdrawal policy on behalf of myself and my co-authors.